



# Performance assessment of the IASI-O3 KOPRA product for observing midlatitude tropospheric ozone evolution for 15 years: validation with ozone sondes and consistency of the three IASI instruments

Gaëlle Dufour[1], Maxim Eremenko[2], Juan Cuesta[2], Gérard Ancellet[3], Michael Gill[4], Eliane Maillard Barras[5], Roeland Van Malderen[6]

[1]Université Paris Cité and Univ Paris Est Créteil, CNRS, LISA, Paris, F-75013, France
[2]Univ Paris Est Créteil and Université Paris Cité, CNRS, LISA, Créteil, F-94010, France
[3]LATMOS, Sorbonne Université, Université Versailles St-Quentin, CNRS/INSU, Paris, France
[4]Met Éireann Valentia Observatory, V227 V23 Cahersiveen, Ireland
[5]Federal Office of Meteorology and Climatology MeteoSwiss, Payerne, Switzerland
[6]Royal Meteorological Institute of Belgium (RMIB), Uccle, Belgium

*Correspondence to*: Gaëlle Dufour (gaelle.dufour@lisa.ipsl.fr)

**Abstract.** The Infrared Atmospheric Sounding Interferometer (IASI) has been monitoring the atmosphere for operational
meteorology and atmospheric composition studies since 2007 with a succession of three instruments aboard the Metop-A (2006-2021), Metop-B (2012-) and Metop-C (2018-) missions. One of the key species monitored is ozone ($O_3$). This study assesses the quality of the regional IASI-O3 KOPRA product, version v3.0, and the consistency of the three IASI instruments, IASI-A, IASI-B, and IASI-C. The IASI-O3 KOPRA products for IASI-A, IASI-B and IASI-C. IASI-B show a very good agreement and consistency, better than 1%, for the tropospheric ozone column (TrOC) and several
partial columns (surface-450hPa, surface-300hPa) over the three domains, Europe, North America, and East Asia of this study. For the quality assessment and trend analyses, we combine the ozone products derived from IASI-A (2008-2018) and IASI-B (2019-2022) without any bias correction. The comparison with homogenized ozone sondes for six northern midlatitude stations reveals a small negative bias of about 3-6% of the IASI-O3 KOPRA products in the troposphere for both profiles and columns with rather good correlation between 0.7 and 0.9 and an error estimate about 15-17% (compared to sondes smoothed with
averaging kernels (AKs)). The ozone variability is also well reproduced for all the partial columns with a slight underestimation of about 10% for the TrOC. Based on the comparison with the ozone sondes, we identified a temporal drift (of about -0.06 ± 0.02 DU/yr in average), while more pronounced in summer, for three different ozone columns (TrOC, surface-450hPa, surface-300hPa). However, a significant variability of the estimated drifts depending on the sample of ozone sonde sites is remarked, that does not allow its use for correcting the IASI ozone product timeseries over broad domains. Whereas the upper
tropospheric ozone trends are mainly positive or undefined, the lower tropospheric ozone trends are mainly systematically negative. The regions the most affected by negative trends are the Mediterranean, Western North America, Eastern North America and East Asia. Compensations between lower and upper tropospheric trends prevent the identification of any specific




long-term behaviour for TrOCs over the three domains. The negative tropospheric ozone column anomalies in the 2020-2022 (post-COVID19) time period observed in our northern hemisphere mid-latitude domains slightly impact the estimated trends

but do not change the conclusions stressed before.

## 1 Introduction

Tropospheric ozone ($O_3$) is a key species for atmospheric chemistry for its decisive role played in the oxidizing capacity of the atmosphere (Monks et al., 2015). In addition to be a short-lived climate forcer (SLCF) and an important greenhouse gas (Szopa et al., 2021), tropospheric ozone is also one of the major air pollutants impacting human health (WMO, 2022), crops and

ecosystems (e.g. Emberson, 2020). As a secondary pollutant, ozone is highly dependent on the emissions of its precursors, such as nitrogen oxides (NOx), non-methane volatile organic compounds (NMVOCs), carbon monoxide (CO) and methane ($CH_4$) (e.g. Atkinson, 2000). With a relatively short lifetime, this leads to a high spatio-temporal variability of $O_3$ distribution at different temporal and spatial scales (Archibald et al., 2020; Cooper et al., 2014; Gaudel et al., 2018; Tarasick et al., 2019; Young et al., 2018).

Monitoring tropospheric ozone distribution and its time evolution is then crucial and part of the Tropospheric Ozone Assessment Report (TOAR) activities (https://igacproject.org/activities/TOAR, last access 16 December 2024). In complement to in situ, ground-based, or aircraft measurements, satellite instruments provide a daily global monitoring capability at high resolution with improved sensitivity to probe the troposphere (e.g. Barret et al., 2020; Boynard et al., 2018; Eremenko et al., 2008; Hayashida et al., 2018; Hubert et al., 2020; Liu et al., 2010; Maratt Satheesan et al., 2024; Miles et al.,

2015; Pope et al., 2023; Ziemke et al., 2019) even towards the lowermost layers (e.g. Cuesta et al., 2013; Fu et al., 2013). However, tropospheric ozone observations from space remains challenging leading to inconsistencies in the spatial and temporal distributions of ozone derived from the different satellite products (e.g. Gaudel et al., 2018). Indeed, Gaudel et al. (2018) showed discrepancies especially for the trends derived from these observations, the ones issued from ultraviolet (UV) sounders suggesting positive recent trends in tropospheric ozone whereas the ones issued from infrared (IR) sounders where

more likely negative. More recent studies show, however, less systematic positive trends derived from the OMI Monitoring Instrument (OMI) for example (Ziemke et al., 2019) and small linear trends with large uncertainties in the lower troposphere using both OMI and the Infrared Atmospheric Sounding Interferometer (IASI) in the most urbanized regions of the northern hemisphere (Pope et al., 2024).

Among the satellite sounders, IASI has been extensively used to trends studies at the global (Wespes et al., 2016, 2017) and

regional scale (Dufour et al., 2018, 2021). The IASI instruments are nadir-viewing Fourier transform spectrometers. They are flying on board the EUMETSAT (European Organisation for the Exploitation of Meteorological Satellites) Metop satellites (Clerbaux et al., 2009). Three versions of the instrument have been operational on the same orbit since 2006: the first one aboard the Metop-A platform between October 2006 and November 2021, the second one aboard the Metop-B platform since September 2012 and still operating, and the third one aboard the Metop-C platform since November 2018 and still operating.



The IASI instruments cover the spectral band between 645 and 2760 cm$^{-1}$ in the thermal infrared, with an apodised resolution of 0.5 cm$^{-1}$. The field of view of the instrument is composed of a $2 \times 2$ matrix of pixels with a diameter at nadir of 12 km each. IASI scans the atmosphere with a swath width of 2200 km and crosses the Equator at two fixed local solar times 09:30 LT (descending mode) and 21:30 LT (ascending mode), allowing the monitoring of atmospheric composition twice a day at any location.

The inconsistencies and large uncertainties on trends estimated from satellite observations and reported in literature (e.g. Gaudel et al., 2018, Pope et al., 2024) stress the need for detailed validation of the satellite observations, including the analyses of possible drifts in the timeseries. This is the objective of the present study to validate and assess the drifts in the IASI-O3 KOPRA product (Eremenko et al., 2008), version 3.0. This version of the product was validated only on the first years of operation of the first IASI instrument aboard Metop-A (Dufour et al., 2021). We extend the validation to a much larger period

(2008-2022) covering almost all the operations of the three instruments aboard Metop-A, Metop-B and Metop-C. We investigate the consistency between the three IASI instruments. As the IASI-O3 KOPRA product is a regional product focusing on urbanized regions of the northern hemisphere (Europe, North America, and East Asia), we validate the product with midlatitudes ozone sondes. We benefit from the homogenization work on the ozone sondes done in the TOAR framework (Van Malderen et al., 2024). Section 2 describes the IASI-O3 KOPRA product and discussed the consistency between the

three IASI instruments. The validation methodology and its results and conclusions are given in Section 3 and the possible consequences of drift correction on trend analyses are discussed in Section 4. A summary and conclusions are displayed in Section 5.

## 2 IASI-O3 KOPRA product

### 2.1 Retrieval

Ozone profiles are retrieved from the IASI L1C radiances using the Karlsruhe Optimized and Precise Radiative transfer Algorithm (KOPRA) radiative transfer model, its inversion tool (KOPRAFIT), and an analytical altitude-dependent regularization method, as described in Eremenko et al. (2008) and (Dufour et al., 2012, 2015). Surface temperature and temperature profiles are retrieved before the ozone retrieval using selected micro-windows in 800-950 cm$^{-1}$ and 670-700 cm$^{-1}$ spectral range respectively. The prior information is from ERA-Interim reanalysis till 2019 (Dee et al., 2011) and then ERA5

reanalysis (Hersbach et al., 2020). Seven micro-windows in the range 975-1100 cm$^{-1}$ are used for the ozone retrieval. Water vapor is fitted simultaneously with ozone to account for interferences in the spectral windows used for the retrieval and to improve the retrieval in the current version (3.0) of the product. The a priori profiles are compiled from the ozone climatology of McPeters et al. (2007). The a priori and the constraints change depending on the tropopause height, taken as the 2 PV geopotential height product from the ECMWF (European Centre for Medium-Range Weather Forecasts). Three cases are

considered with specific a priori and constraint for each of them: the polar case when tropopause height is lower to 10 km, the midlatitude case when tropopause height is within 10-14 km and the tropical case when tropopause height is larger than 14





km. A data screening procedure is applied to filter cloudy scenes and to ensure the data quality (Dufour et al., 2010, 2012; Eremenko et al., 2008). The retrieval is performed for the morning pixels of three geographical regions of the northern hemisphere: Europe (35°N-70°N, 15°W-35°E), North America (25°N-60°N, 70°W-125°W), and East Asia (20°N-55°N, 100°E-150°E), named, respectively, EU, US, and CH in the following. The data were processed for the three IASI instruments between 2008 and 2020 for IASI aboard Metop-A (named IASI-A), between 2013 and 2023 for IASI aboard Metop-B (named IASI-B) and between 2019 and 2023 (named IASI-C). In the following, we limit the comparisons and analyses to the 2008-2022 period to be consistent all over the sections. IASI-A data are considered only till 2018, as after this year, some end-of-life technology test campaigns were conducted on the Metop-A payload (Tarquini, 2018). As level 1C delivery of IASI-C spectra started around April, we consider IASI-C data only from 2020 to cover full years. The trend analyses are then based on IASI-A from 2008 to 2018 and IASI-B from 2019 to 2022. In the following, retrieved profiles but also different partial columns are considered. The tropospheric ozone column (TrOC) is calculated from the surface to the tropopause given by the WMO lapse-rate definition. We use the ERA5 tropopause height from the Reanalysis Tropopause Data Repository (Hoffmann and Spang, 2022b) at 1°x1° resolution (Hoffmann and Spang, 2022a; Zou et al., 2023). Ozone partial columns from the surface to 300 hPa as recommended by the TOAR for midlatitudes are considered as well as the partial columns from the surface to 450 hPa to avoid contamination from the stratosphere due to the limited vertical resolution of IASI retrievals.

## 2.2 Consistency between IASI-A, IASI-B, and IASI-C

The IASI-B instrument has a large period of operation in common with IASI-A (2013-2018) and with IASI-C (2020-2022), and IASI-A and IASIC do not have common period of operation with high quality data. We then use IASI-B as the reference to study the consistency between the tropospheric ozone retrievals from the three instruments. Table 1 shows the normalized mean bias between IASI-A and IASI-B and between IASI-C and IASI-B for the three regions (EU, US, CH) and three partial columns (TrOC, surface-450hPa and surface-300hPa). A small negative bias, smaller than 1%, compared to IASI-B is observed for both IASI-A and IASI-C for all the columns and regions. This is consistent with results shown by (Boynard et al., 2018) for the FORLI IASI $O_3$ product. The tropospheric ozone columns (TrOCs) show the smallest bias with errors (standard deviation) of the order of or larger than the bias itself for all the regions. The bias is slightly larger for the surface-450hPa and surface-300hPa columns with the standard deviation ranging from 40% to 100% of the bias depending on the region and the column. The mean bias calculated for the European region tends to be the largest. The bias between IASI-C and IASI-B is also larger than the one between IASI-A and IASI-B and the standard deviation smaller. This can be partly explained by the difference between the length of the time periods considered for the comparison.



**Table 1: Normalized mean biases and standard deviations (%) between IASI-A and IASI-B for 2013-2018 (AB) and between IASI-C and IASI-B for 2020-2022 (BC) for the three domains and for three partial ozone columns. IASI-B is taken as the reference**

|  | | TrOC | 450 hPa | 300 hPa |
|---|---|---|---|---|
| EU | | AB: -0.53 ± 0.36 | AB: -0.87 ± 0.47 | AB: -0.75 ± 0.48 |
|  | | BC: -0.48 ± 0.45 | BC: -0.96 ± 0.36 | BC: -0.78 ± 0.36 |
| US | | AB: -0.32 ± 0.43 | AB: -0.62 ± 0.52 | AB: -0.55 ± 0.58 |
|  | | BC: -0.40 ± 0.36 | BC: -0.79 ± 0.33 | BC: -0.63 ± 0.32 |
| CH | | AB: -0.17 ± 0.46 | AB: -0.73 ± 0.48 | AB: -0.63 ± 0.47 |
|  | | BC: -0.43 ± 0.42 | BC: -0.68 ± 0.33 | BC: -0.60 ± 0.30 |

The analysis of the TrOC monthly timeseries averaged over each domain (Fig. 1) confirms the overall good agreement between the three IASI instruments. Seasonal variability derived from the three instruments is very similar and the same drop since 2020 is observed by IASI-B and IASI-C. It is worth noting that this drop in 2020 and 2021 is also reported for the OMI-MLS ozone product (Ziemke et al., 2022) and is ascribed there to the reduced emissions of ozone precursors across the Northern Hemisphere due to COVID-19 lockdown restrictions. The timeseries of the differences between IASI-A, IASI-C, and IASI-B (right panels of Fig. 1) show that the differences can vary at the scale of month or year preventing assessing a systematic bias between the instruments. Similarly, the spatial distributions of TrOC are in very good agreement between the different instruments (Fig. 2 and Fig. B1) and the differences are not systematic enough to derive a systematic bias correction to be apply to all the IASI-A or IASI-C pixels. Similar results are obtained for surface-450hPa and surface-300hPa columns (not shown). As trends are calculated merging IASI-A and IASI-B over 2008-2022 in the following, we checked the impact of correcting or not the bias between the two instruments for the trend calculation of the surface-450hPa and surface-300hPa in Europe where the biases are the largest (Table 1). The trends are calculated according to the method described in Appendix A. The trend without bias correction is -0.07±0.02 DU/yr and -0.10±0.03 DU/yr for surface-450hPa and surface-300hPa columns respectively and with bias correction, -0.08±0.01 DU/yr and -0.13±0.03 DU/yr respectively. The derived trends remain in agreement, within their uncertainties, whether bias correction is applied or not. According to these results, the biases are considered as negligible, and no bias correction is applied in the following.



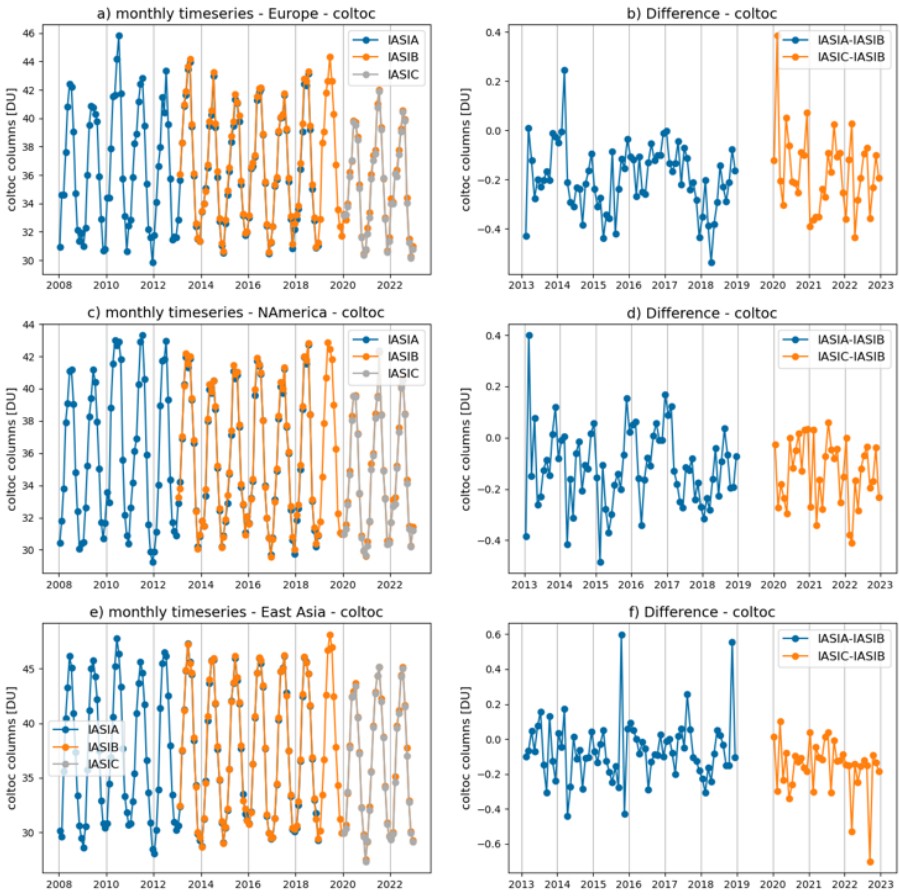

**Figure 1: (left) Monthly timeseries of tropospheric ozone columns derived from IASI-A, IASI-B, and IASI-C; (right) Temporal differences between IASI-A and IASI-B and between IASI-C and IASI-B.**



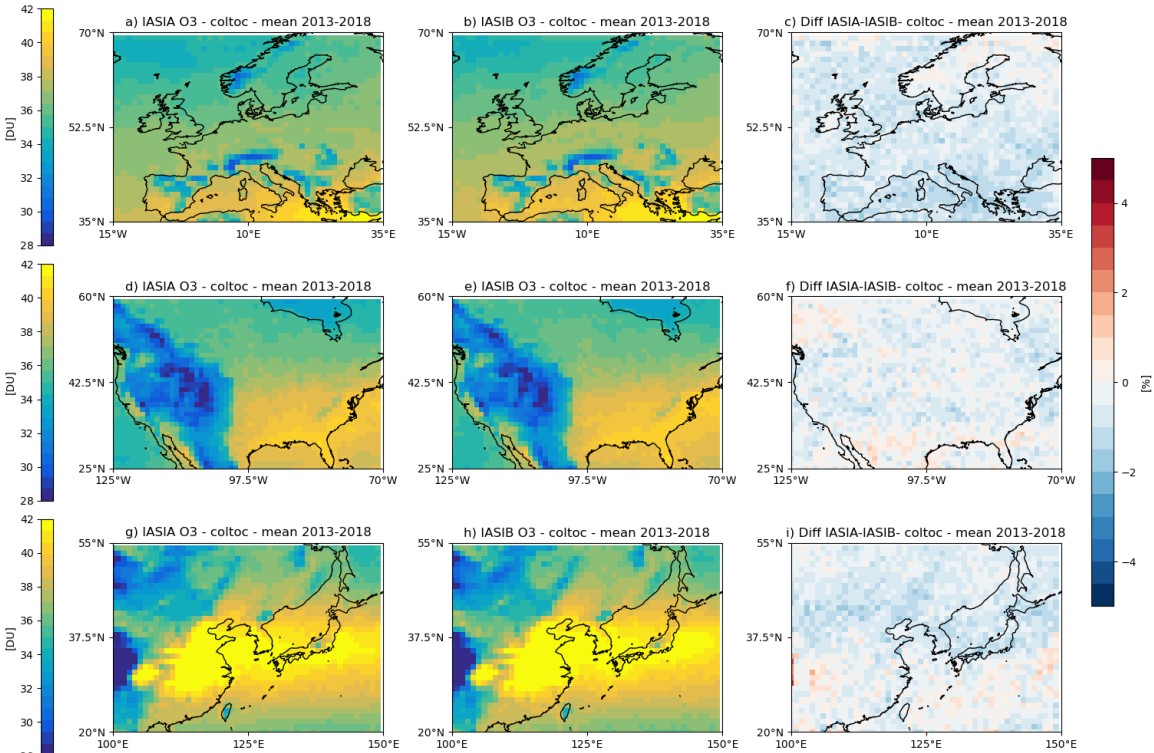

Figure 2: **Mean ozone distribution of tropospheric ozone columns derived from IASI-A (left) and IASI-B (middle) for 2013-2018 at 1°x1° resolution and their relative differences (right) over the three regions.**

# 3 Validation against ozone sondes

## 3.1 Ozone sondes description

We use the dataset of ozone sonde time series homogenized in the framework of the HEGIFTOM (Harmonization and Evaluation of Ground-based Instruments for Free-Tropospheric Ozone Measurements) Focus Working Group of the TOAR-II IGAC initiative as reference for the IASI-O3 KOPRA v3 validation (https://hegiftom.meteo.be, last access 16 December 2024). All the ozone sondes timeseries are corrected from possible biases related to instrumental or processing changes in this dataset (Van Malderen et al., 2024 and references therein). As the IASI-O3 KOPRA product is available only for three regions (Europe, North America and East Asia), including mainly midlatitudes, we focus the comparison with the sites lying in the 30°N-60°N latitude band. We considered only the subset of homogenized sondes covering the entire period from 2008 to 2022 and matching the coincidence temporal criteria described in Section 3.2. We made a selection of six stations for the comparison, five over Europe (Legionowo, OHP, Payerne, Uccle, Valentia) and one over North America (Boulder). Figure 3 displays the location of the sondes as well as the number of coincident days of measurements between sondes and IASI observations. In



total, 5966 sondes profiles are used for the validation. Payerne and Uccle stations provide the largest number of profiles,
counting for more than 60% of the total sonde profiles.

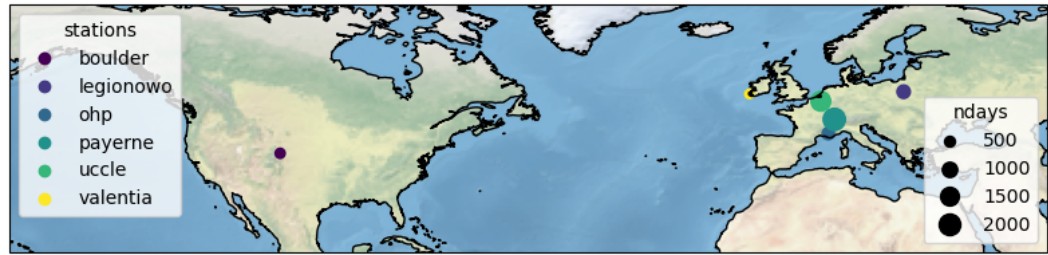

**Figure 3: Location of the ozone sonde stations used for the validation of the IASI-O3 KOPRA product and number of days of sonde measurements considered over the 2008-2022 time period.**

### 3.2 Methodology

The coincidence criteria used for the validation are ±1° in latitude and ±1° in longitude around the sonde station, a time difference shorter than ± 6 h, and a minimum of 10 clear-sky IASI pixels matching these criteria. These coincidence criteria are in the range of criteria reported in literature for IASI ozone validation (Barret et al., 2020; Boynard et al., 2016, 2018; Dufour et al., 2012, 2015, 2018). We use the validation method described e.g. by Dufour et al. (2012) and consider both the sonde profiles smoothed by the averaging kernels (AKs) of each pixel and the raw sonde profiles for the comparison as done by Dufour et al. (2012) and recommended by Barret et al. (2020). To smooth the sonde profiles by the IASI AKs, we need to complete the sonde profile up to 60 km, the altitude range of the IASI product. As the top altitude of the sonde profiles is variable and as we are mainly interested by the troposphere, we decided to complete the sonde profiles with the a priori profiles from 20 km to 60 km altitude to maintain a certain homogeneity in the sonde treatment. For comparison and AKs smoothing, we need to use the IASI vertical grid (typically 1 km resolution in the troposphere) and regrid the sonde profile to this grid. To avoid possible interpolation issues and uncertainties, we convert the volume mixing ratio (vmr) profiles and the AK matrices into the partial columns space before smoothing and comparison following Keppens et al. (2015). The resulting profiles are then converted back to vmr profiles. The validation is done on both the profiles and several partial columns. In addition to TrOC, surface-450hPa, surface-300hPa and 450hPa-tropopause columns, we also present results for the surface-6km and 6-12km columns for comparison with previous validations of the IASI-O3 KOPRA products (Dufour et al., 2012, 2015).

The validation is performed for 2008-2022 merging IASI-A (2008-2018) and IASI-B (2019-2022) datasets. Global statistics over the period, such as mean biases, root mean square of errors (RMSE) and correlations, are calculated for both profiles and columns. A temporal analysis is also performed to identify possible drifts in the IASI data.



### 3.3 Results

### 3.3.1 Profiles analyses

Figure 4 compares the mean vertical profile of ozone measured by the sondes to the mean IASI-O3 KOPRA profile and the mean a priori profile. Both the raw and smoothed profiles are displayed for the sondes. As expected, the limited vertical resolution of the IASI profile and the a priori contribution to the retrieval influence the shape of the vertical profile as shown by the difference between the raw and smoothed profiles. The ozone vmrs in the free troposphere are reduced by about 10% when the sondes are smoothed and the height of the ozonopause (where the gradient of the ozone profile is the maximum)

seems strongly affected by the shape of the a priori profile. This leads to oscillations between -20% to 20% in the normalized mean bias profile between 9 to 15 km. Outside this range, the bias is rather constant in altitude, especially in the free troposphere where a negative bias of about 15% between the raw profile and the IASI profile is observed. When the sonde profile is smoothed by the AKs, the smoothing errors are removed, and the smoothed sonde profile and the IASI profile compare better in shape and in magnitude. The normalized mean bias in the free troposphere is about -6%, IASI vmrs being

smaller. A smoothed change between a negative and positive bias operates in the upper troposphere – lower stratosphere region. The normalized RMSE, which gives an estimate of the observation errors (Dufour et al., 2012), is slightly less than 20% in the troposphere compared to the smoothed sondes. When considering the smoothing errors (comparison to raw profile), the normalized RMSE is about 28% in the free troposphere. The largest differences and RMSE in the first two kilometers when comparing to raw sondes are likely due to an issue because high altitude stations (Payerne, OHP, and Boulder) are mixed

with low altitude stations. It worth noting that the better agreement with smoothed sondes is due by the lack of sensitivity of IASI observations at the surface. Then, most of the information comes from the a priori in both IASI and smoothed profiles.

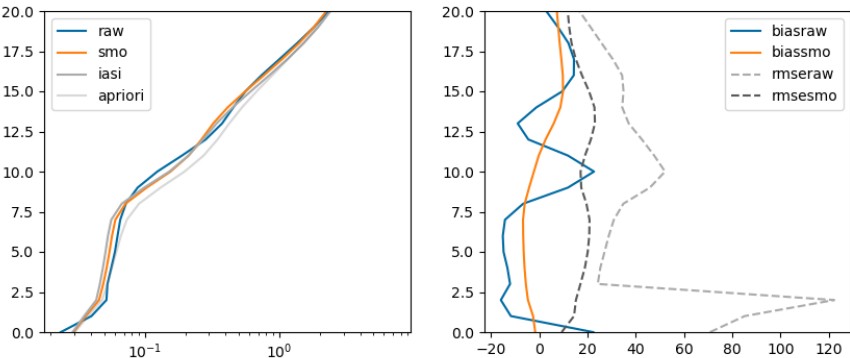

**Figure 4: (left) Mean vertical profiles of ozone for the raw and smoothed sonde and IASI in coincidence (see text for coincidence criteria). The mean a priori profile used for the retrievals is displayed. The mean is calculated over 2008-2022 and includes IASI-A**

**(2008-2018) and IASI-B (2019-2022) data. (right) Mean bias and RMSE profiles of IASI ozone profiles against raw and smoothed sondes profiles.**

Besides the mean profiles, we also analyze the temporal evolution of the monthly profiles, especially their anomalies. Figure 5 shows the curtain plots of this evolution. To calculate the anomalies, the mean seasonal cycle of the profiles is subtracted





from the monthly profiles. The anomalies are normalized by the standard deviation of the mean profiles to calculated a
normalized deviation (Chang et al., 2022). The main large patterns of the anomalies are rather well consistent in the free
troposphere between IASI, raw and smoothed sonde anomaly profiles with mainly positive anomalies between 2008 and 2010,
negative anomalies in 2011/2012, and negative anomalies after 2020. These latter are in agreement with other datasets and
studies and are ascribed to precursor emission reduction during COVID-19 pandemic (e.g. Chang et al., 2022; Steinbrecht et
al., 2021; Ziemke et al., 2022). Figure 5 also illustrates well the limited vertical resolution of IASI, visible in both the IASI
and smoothed sonde anomaly profiles. For example, the negative anomaly in 2011 located above 7 km in the raw sonde
anomaly profile extends down to the lower free troposphere in the IASI and smoothed sonde anomaly profiles and similarly
for the positive anomaly in 2013. In the case where variabilities in altitude are visible in the raw sonde anomaly profiles such
as in 2015, the IASI anomaly profiles are more homogeneous in altitude than the smoothed sonde profiles. This suggests that
the limited vertical resolution of IASI, which is transposed into the sonde profiles when smoothed by the AKs, does not fully
explain the difficulties to reproduce these vertical variabilities with IASI. More generally, the agreement between IASI, raw
and smoothed sonde anomaly profiles is more variable over the 2013-2019 period, when the raw sonde anomaly profiles show
larger variabilities in altitude. The main discrepancies between the IASI and sonde anomaly profiles appear in 2012 where
IASI shows a large positive anomaly and in 2018 and winter 2021/2022 where the sondes show positive anomalies but not
IASI. In 2019, the negative anomaly is rather consistent with the raw sonde anomaly profile but not the smoothed one, where
it is not visible. Overall, these comparisons show a tendency towards more negative anomalies at the end of the period
associated to COVID-19 pandemic. As this effect seems to be more pronounced for IASI, the contribution from a possible
drift in the IASI data cannot be ruled out. This will be analyzed with the partial columns in the next sections.

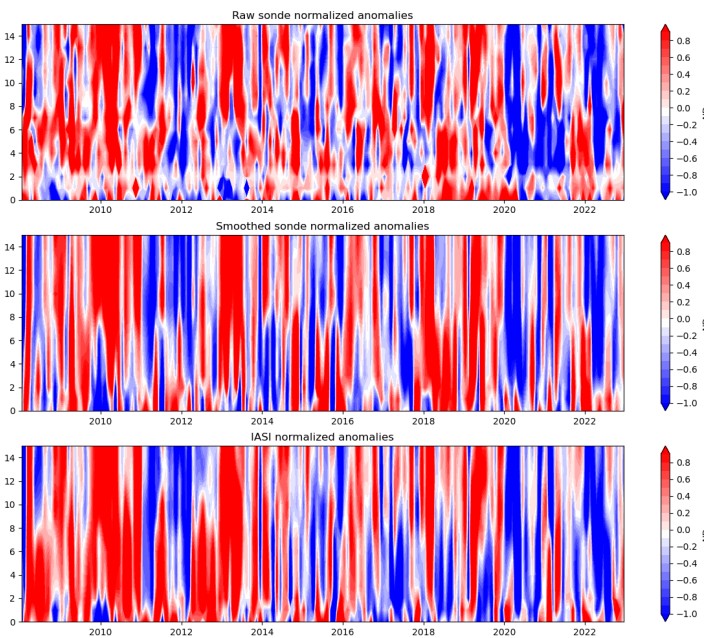



**Figure 5: Temporal evolution of monthly anomaly profiles between 2008 and 2022. The anomaly profiles are normalized by the**
**standard deviation of the profile. The top panel displays the raw sonde profiles, the middle panel the smoothed sonde profiles, and**
**the bottom panel the IASI profiles**

### 3.3.2 Global statistics on the partial columns

Figure 6 shows the global statistics over the entire period (2008-2022) of the comparison between IASI and sonde tropospheric
ozone columns. The statistics are calculated using over the daily averages and both for all the stations together and individually.
Both the comparison to raw (stars) and smoothed (circles) sondes are shown. The results for the other partial columns (surface-
450hPa, surface-300hPa, 450hPa-tropopause, surface-6km, and 6-12km) are given in Appendix C. The normalized mean bias
of the TrOC retrieved from the IASI-3 KOPRA product is slightly negative, -0.7 % and -3.3%, compared to raw and smoothed
sonde TrOC respectively. It varies between -7% for OHP and 2.4% for Payerne for the comparison with raw sondes and from
-9.4% for OHP to -0.1% for Legionowo for the comparison with the smoothed sondes. The normalized mean bias is slightly
larger (~ -5.5%) for partial columns in the troposphere when compared to smoothed sondes (Appendix C). It reaches between
-10% and -13% when compared to raw sondes. The smaller biases assessed for the TrOC compared to the lower troposphere
is likely explained by the smallest difference between IASI and the sonde in the upper troposphere (Figs. C3 and C4). Negative
and positive differences partly compensate as seen in the vertical profiles (Fig. 4). In terms of correlation, standard deviation
and RMSE, the Taylor diagram (Fig. 6) shows that these statistical indicators improve when comparing with smoothed sondes
which is expected as the smoothing errors is removed in this case. The RSME displayed in the Taylor diagrams (curved lines
in Fig. 6, Appendix C) are normalized against the standard deviation of the sondes. If we calculate them as a percentage of the
columns, they give an estimate of the column's uncertainties. They range from 15-17% for the estimation against smoothed
sonde columns and 20-25% for the estimation against raw sonde columns. Correlations larger than 0.8 are obtained for TrOC
when compared to smoothed sondes (Fig. 6) but they are smaller for partial columns including the lower troposphere around
0.7 and around 0.9 for upper tropospheric – lower stratospheric columns. This difference between lower tropospheric and
upper tropospheric – lower stratospheric columns might be explained by the better sensitivity of IASI in the free to upper
troposphere. In addition, the variability of the 6-12km and 450hPa-tropopause columns estimated by their standard deviation
(about 40% and 35% respectively) is larger compared to the one in the surface-450hPa column (about 20%). This larger
variability is mainly associated to large scale dynamical processes which affect the tropopause height. These large-scale
modulations are usually well captured with IASI (Dufour et al., 2015) and mainly drive the high correlation observed. In the
lower troposphere, the variability is smaller and of the order of the errors on the IASI columns, the correlation performances
calculated on daily data are then more affected by noise. One can also notice that the variability of the TrOC is slightly
underestimated by IASI (by about 10%) (Fig. 6). Globally, the performances of the IASI-O3 KOPRA product v3.0 are similar
to what was reported in previous validations for other product versions (Dufour et al., 2012, 2015, 2018, 2021).






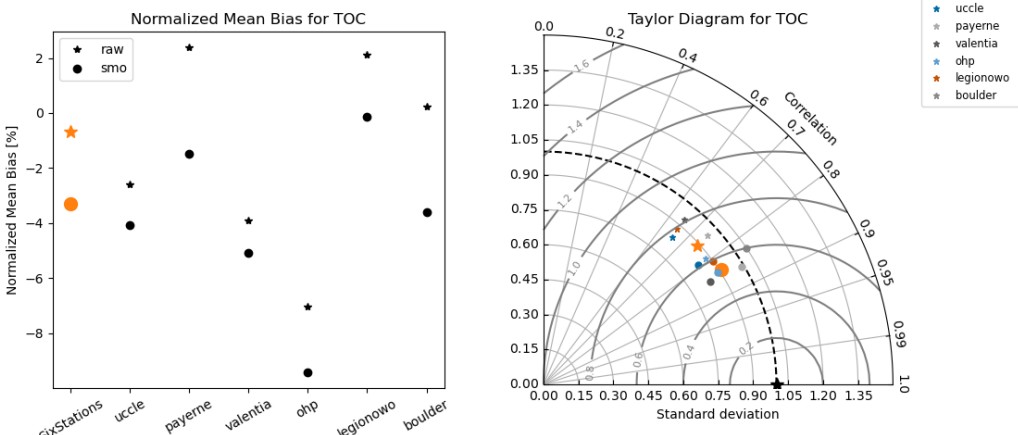

**Figure 6: (left) Normalized Mean Bias (NMB) of the IASI TrOC against raw and smoothed sondes. The NMB is given for the individual sonde stations and globally for the six stations together. (right) Taylor diagram for the tropospheric ozone columns (TrOC) including also statistics against raw (stars) and smoothed (circles) sondes for individual and the six stations together. Curved lines denote the RSME, normalized against the standard deviation of the sondes.**

### 3.3.3 Drift analysis


The analysis of the temporal evolution of the vertical profiles and their anomalies (Section 3.3.1) suggests that a possible drift in the IASI observations might appear with time in the troposphere. As the vertical resolution of IASI is limited, we evaluate and quantify the possible drift on tropospheric and partial tropospheric columns in this section. The methodology to calculate the drift is described in Appendix A. The drift is calculated against both the smoothed and the raw sondes columns. We recall

here that monthly timeseries are used for this estimation.

First, we analyze the monthly timeseries of the smoothed sondes and IASI for the TrOC and for lower (surface-450hPa) and upper (450hPa-tropopause) tropospheric columns (Fig. 7). A very good agreement is observed between the sonde and IASI timeseries for the upper troposphere. The agreement in the lower troposphere is less good and gradually deteriorates with time, especially in summer and after 2014. The amplitude of the seasonal cycle is underestimated with IASI and its maximum is

shifted towards spring. This deterioration of the agreement is also visible in TrOC but to a lesser extent, indicating some compensations between the lower and upper troposphere which limit the impact on the TrOC.



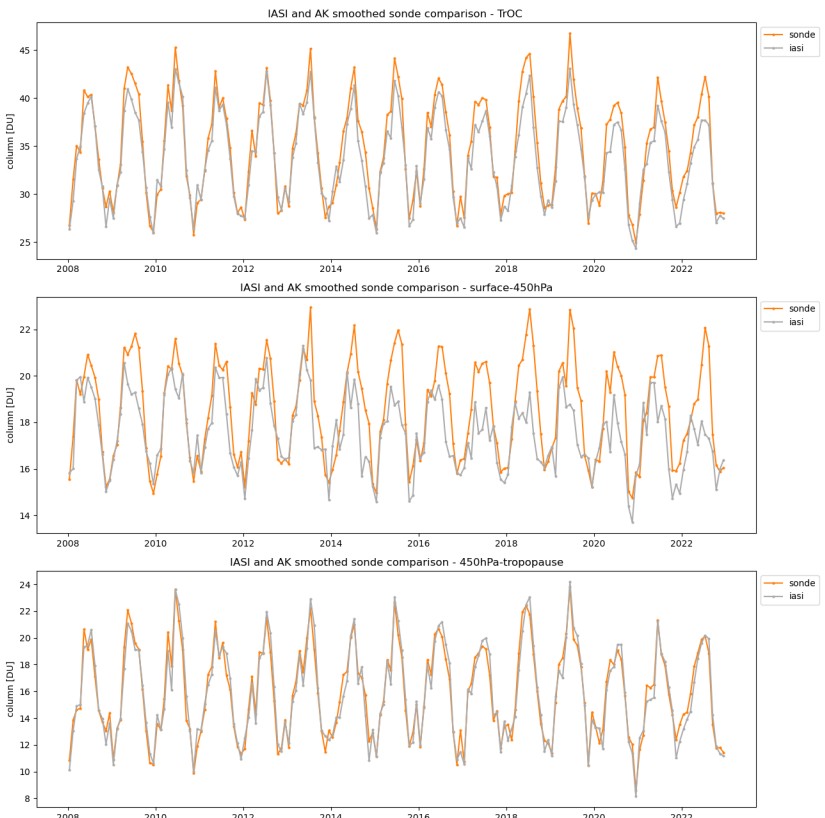

**Figure 7: Monthly timeseries from 2008 to 2022 of the TrOC (top), surface-450hPa (middle), and 450hPa-tropopause (bottom)**
**columns for smoothed sondes (orange) and IASI (grey).**

Table 2 summarizes the drifts derived for TrOC and different partial columns. An overall negative drift is estimated for all the

columns both from the smoothed and the raw sondes. The drift is about -0.06 DU/yr against the smoothed sondes for columns

including the lower troposphere (TrOC, surface-450hPa, surface-300hPa) and more variable for the ones derived from the raw

sondes. The p-value associated to these drift estimates is smaller than or equal to 0.05 suggesting that the drift might be

considered and corrected when analyzing timeseries, especially for trend studies. As shown in Fig. 7, the discrepancies between

sondes and IASI are mainly in summer. The seasonal drifts calculated by a simple linear regression are higher and more

significant in summer (-0.15 DU/yr, p=0.009 for TrOC) than for other seasons (-0.03 DU/yr, p=0.47 for winter, -0.06 DU/yr,

p=0.05 for spring, -0.09 DU/yr, p=0.15, for TrOC). The partial column in the upper troposphere (450hPa to tropopause) show

much smaller drifts, insignificant, with large uncertainties and p-values (Table 2). These results suggest the TrOC drift is

driven by the lower troposphere. We did several tests to try to identify the reasons of these drifts. We evaluated the sensitivity

of the retrieval to the temperature and humidity profiles, to the definition of the tropopause height used to identify the retrieval

case (polar, midlatitudes or high latitudes) and changing the a priori profile used but no one of these tests were conclusive to

explain or remove the drift, especially during summer. We can mention that the quality of the spectra fit, given by the root



mean square error of the fit, reduces slightly with time. However, the increase in the RMSE of the fit affects rather similarly retrievals in summer and winter (+6% and +5% of increase between the beginning and the end of the period respectively) and would not explain the larger summer drift. Looking at the thermal contrast (the difference between surface temperature and air temperature just above) shows that the mean thermal contrast for the pixels coincident with the subset of the six sondes used is mainly negative and its absolute value tends to become closer to zero with time, especially in summer (not shown). This

could explain a loss of sensitivity in the lower troposphere with time. However, the smoothing of the sonde profiles with the AKs should integrate this loss of sensitivity which should not explain the summer differences between smoothed sondes and IASI. We also investigated the impact of the spatial coincidence criterium reducing it to 0.5° or increasing it to 2° around the site but summer differences remain.

**Table 2: Drifts, in DU/yr, derived from the comparison of IASI-O3 KOPRA products to smoothed and raw sondes for the 2008-2022 period for columns including the lower troposphere (TrOC, surface-450hPa, surface-300hPa) or not (450hPa-tropopause, and 300hPa-tropopause columns).**

|  | Drift against smoothed sondes | Drift against raw sondes |
|---|---|---|
| TrOC | -0.057 ± 0.029 (p=0.05) | -0.088 ± 0.030 (p=0.004) |
| 450 hPa | -0.062 ± 0.018 (p=0.000) | -0.051 ± 0.026 (p=0.05) |
| 300 hPa | -0.072 ± 0.025 (p=0.005) | -0.097 ± 0.037 (p=0.01) |
| 450 hPa - tropopause | -0.012 ± 0.013 (p=0.34) | -0.012 ± 0.017 (p=0.47) |

The drift estimation in Table 2 integrates the six stations used for the validation but we also analyzed the results station by

station. Figure 8 shows the individual drift of IASI-O3 KOPRA product for each of the six stations for the TrOC against both smoothed and raw sondes. The drift is largely variable ranging from +0.01 DU/yr (p=0.85) for Uccle to -0.25 DU/yr (p<0.001) for OHP for example when calculated against smoothed sondes. It is worth noting that the Uccle and Payerne stations are the ones with the largest number of coincident profiles (Fig. 3), so they should influence more strongly the drift calculations than the other stations. This variability of the drift between the stations might question how much one can trust the value of the drift

derived for the six stations to use it to correct the drift and finally how much the sonde stations are representative of drift over a larger domain such as entire Europe, US or China. Also, five of the six stations are in Europe, only one in the US and zero in East Asia, questioning the representativeness and robustness of the drift correction one can derive.





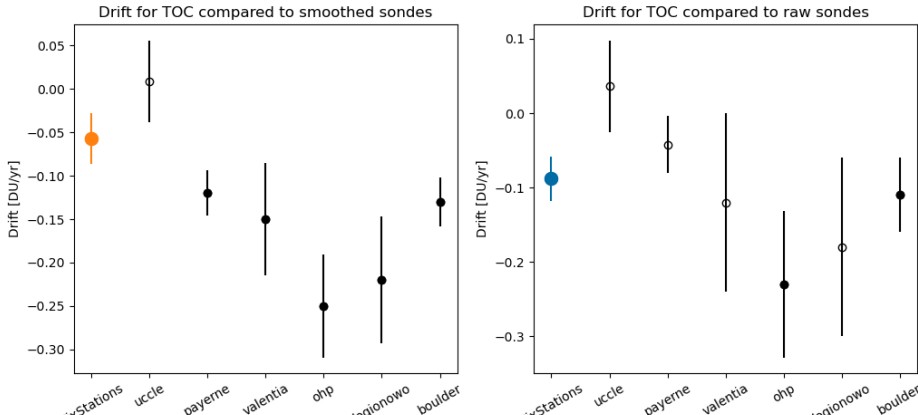

**Figure 8: Drifts of the IASI-O3 KOPRA product derived from the comparison with smoothed (left) and raw (right) sondes for the**
**TrOC. The individual drifts by station and the overall drifts estimated from the six stations are indicated. Open circles correspond to drifts associated with a p-value larger than 0.05.**

## 4 Discussion on recent trends

In this section, we discuss the trends derived from IASI-O3 KOPRA product. Figure 9 shows the 1°x1° trends of the TrOC over the three domains. Figures D1 and D2 shows them for the lower and upper tropospheric columns. The trends are calculated
according to the methodology described in Appendix A. The p-value is also reported. In the lower troposphere, the trends are unambiguously negative for all the domains with p-value almost systematically smaller than 0.05 (yellow color) (Fig. D1). Stronger negative trends are observed over the Mediterranean in Europe, over western US and east of Florida in North America, and over the North China Plain, northern China and the downwind regions towards Japan in East Asia. On the contrary, the trends with p-values smaller than 0.05 in the upper troposphere are positive (Fig. D2). This is especially the case for the East
Asia domain where most of Central East China and regions below 35°N show large positive trends in the upper troposphere. The differences between trends in the lower and upper troposphere lead to more contrasted trends for the TrOC which combine both the lower and upper troposphere (Fig. 9). The trends are mainly negative south to 60°N in Europe, for all the domain in North America, and north to 30°N in East Asia. The p-value of the most negative trend is generally smaller than 0.05.



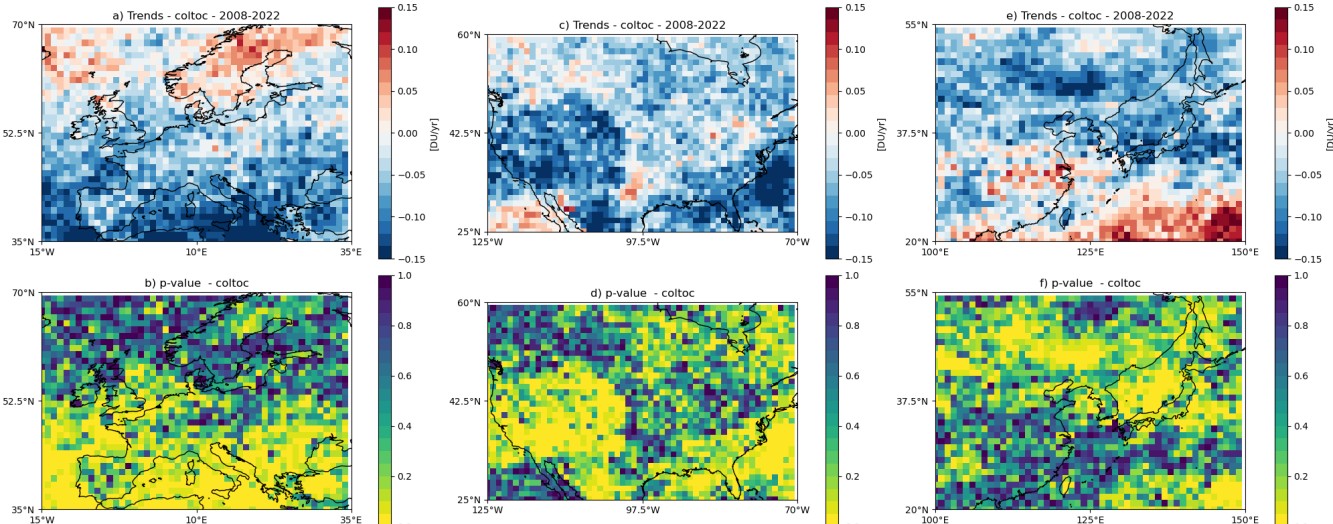

**Figure 9: TrOC trends between 2008 and 2022, in DU/yr, estimated from the IASI-O3 KOPRA product for the three domains, Europe (first column), North America (second column), and East Asia (third column). The corresponding p-values are displayed in the second row. The yellow color indicates p-values smaller than 0.05.**

We also calculated regional trends over the three domains and using the regions defined by Iturbide et al. (2020) for the Sixth IPCC assessment report (AR6) (Table 3). As for the gridded trends, the surface-300hPa and surface-450hPa columns show persistent negative regional trends with small p-values (<0.05) for all the regions. It is worth noting that Pope et al. (2024) also identified negative trends for the surface-450hPa columns over Europe, North America and East Asia within OMI and IASI (FORLI product, Boynard et al., 2018) datasets for 2008-2017 but with large uncertainties. On the contrary, the upper tropospheric regional trends are mainly positive but insignificant as the uncertainties and the p-values are large, except in North Central America and East Asia. Likely due to the compensation of the lower and upper tropospheric trends, the regional TrOC trends are overall negative, although associated with large uncertainties and p-values. The TrOC regional trends are nevertheless more significantly negative when estimated over the entire Europe, the Mediterranean, and Western North America. For these regions, the trends range from -0.05 DU/yr to -0.11 DU/yr and are likely driven by the stronger lower tropospheric trends in these regions. As the retrievals from infrared sonders such as IASI are not very sensitive to the lowermost troposphere with potentially a large contribution of the a priori, we checked if a negative trend is present in the a priori surface-300hPa and surface-450hPa partial columns which could explain the observed negative trends for these columns. The trends for these a priori columns are rather homogenous over the domains and regions: positive, about $0.02 \pm 0.01$ DU/yr, with quite small p-values (mainly <0.1) for the different IPCC regions in Europe and North America and no noticeable trend identified for East Asia. The negative trends reported in the lower troposphere are then not affected by the a priori. Despite the uncertainties remaining on the representativeness and robustness of the drifts estimated in Section 3, we corrected the TrOC and lower tropospheric IASI timeseries for these drifts and evaluated the resulting trends for illustration purposes only. When the drift is corrected, no specific trends (large uncertainties and p-values) are estimated for the TrOC. The negative trends in





the lower troposphere are not systematic then. They persist in the Mediterranean, Western North America, Eastern North America and East Asia.

Finally, as the end of 2008-2022 is affected by the COVID-19 pandemic and the associated reduction of ozone precursor emissions, we also evaluate the trends for 2008-2019 for comparison (Table D1). The lower tropospheric trends remain negative with p-values smaller than 0.05, except for Northern Europe and North Central America where the p-values are larger. The trends are similar or only slightly smaller than the trends for 2008-2022 suggesting that the negative trend in lower tropospheric ozone was already well established before the additional reduction of lower tropospheric ozone due to COVID-

19 lockdowns. The upper tropospheric trends become systematically positive with p-values smaller than 0.05 for 2008-2019 whereas they were mainly positive but insignificant for 2008-2022. This suggests that the COVID-19 period affects ozone distributions and trends up to the upper troposphere. Finally, when excluding the COVID-19 period, the compensation between the lower and upper tropospheric ozone behavior is more visible for 2008-2019 and is reflected in TrOC for which mainly no specific trends are observed (Table D1). The only exceptions are for Northern Europe and East Asia when the TrOC trends are

positive (0.09 DU/yr and 0.14 DU/yr respectively).

**Table 3: Trends between 2008 and 2022, in DU/yr, calculated for different regions defined for the Sixth IPCC assessment Report and the three main domains. The corresponding p-values are indicated in parenthesis. The trends are provided for four partial columns: TrOC, surface-300hPa, surface-450hPa, and 450hPa-tropopause. The trend values are bolded when the p-value is smaller or equal to 0.05.**

| Trends | TrOC | 300 hPa | 450 hPa | 450hPa-tpp |
|---|---|---|---|---|
|  | DU/yr | DU/yr | DU/yr |  |
| **Europe** | **-0.05 ± 0.02 (p=0.03)** | **-0.12 ± 0.03 (p<0.001)** | **-0.07 ± 0.02 (p<0.001)** | +0.01 ± 0.02 (p=0.73) |
| WCE | -0.05 ± 0.04 (p=0.22) | **-0.10 ± 0.03 (p<0.001)** | **-0.06 ± 0.02 (p=0.005)** | -0.01 ± 0.04 (p=0.86) |
| NEU | -0.02 ± 0.04 (p=0.56) | **-0.09 ± 0.02 (p<0.001)** | **-0.06 ± 0.01 (p<0.001)** | +0.02 ± 0.03 (p=0.47) |
| MED | **-0.11 ± 0.04 (p=0.01)** | **-0.16 ± 0.02 (p<0.001)** | **-0.12 ± 0.02 (p<0.001)** | -0.01 ± 0.04 (p=0.72) |
| **North America** | -0.03 ± 0.03 (p=0.34) | **-0.12 ± 0.02 (p<0.001)** | **-0.08 ± 0.01 (p<0.001)** | +0.03 ± 0.03 (p=0.40) |
| WNA | **-0.10 ± 0.05 (p=0.05)** | **-0.17 ± 0.03 (p<0.001)** | **-0.10 ± 0.02 (p<0.001)** | +0.01 ± 0.03 (p=0.76) |
| CNA | -0.05 ± 0.04 (p=0.17) | **-0.11 ± 0.02 (p<0.001)** | **-0.08 ± 0.01 (p<0.001)** | +0.05 ± 0.04 (p=0.20) |
| ENA | -0.05 ± 0.04 (p=0.14) | **-0.14 ± 0.02 (p<0.001)** | **-0.09 ± 0.01 (p<0.001)** | +0.02 ± 0.03 (p=0.57) |
| NCA | -0.05 ± 0.04 (p=0.26) | **-0.10 ± 0.04 (p=0.01)** | **-0.08 ± 0.02 (p<0.001)** | **+0.07 ± 0.03 (p=0.01)** |
| **East Asia*** |  |  |  |  |
| EAS | -0.05 ± 0.03 (p=0.16) | **-0.15 ± 0.03 (p<0.001)** | **-0.10 ± 0.01 (p<0.001)** | **+0.07 ± 0.03 (p=0.04)** |

WCE: Western and Central Europe, NEU: Northern Europe, MED: Mediterranean, WNA: Western North America, CNA: Central North America, ENA: Eastern North America, NCA: North Central America, EAS: East Asia

* As the East Asia domain of our study is close to the EAS region defined for the Sixth IPCC assessment report, we provide only trends for the latter.



## 4 Conclusion

The aim of this study was to assess the quality of the IASI-O3 KOPRA product, version v3.0, applied to the three IASI
instruments and provide recommendations for its use in trend analyses. The evaluation is done over the domains of the northern
hemisphere where the product is available (Europe, North America and East Asia). First, we assessed the consistency of the
IASI-O3 KOPRA product between the three IASI instruments, IASI-A, IASI-B and IASI-C. IASI-B is considered as the
reference for the comparison and we showed that the three instruments are in very good agreement, better than 1%, for several
partial columns in the troposphere and in the three domains. Our tests showed that a bias correction is not necessary to combine
the different IASI instruments for time series analysis. Therefore, we used a combination of IASI-A (2008-2018) and IASI-B
(2019-2022) without bias correction in this study.

We assessed the quality of the IASI-O3 KOPRA product by comparing with ozone sondes for six northern midlatitude stations
for profiles and different partial columns (surface-300hPa, surface-450hPa, 450hPa-tropopause, surface-6km, 6-12km) and the
TrOC. A small negative bias of about 3-6% in the troposphere is identified when IASI profiles and columns are compared to
sonde profiles and columns smoothed by the AKs of IASI. Correlations between 0.7 and 0.9 are observed depending on the
partial columns considered and errors about 15-17% (compared to smoothed sondes) are estimated. The ozone variability is
well reproduced for all the partial columns with a slight underestimation of about 10% of this variability for the IASI TrOC
compared to ozone sondes.

Based on the comparison with those six ozone sondes time series, we identified a possible drift with time in the ozone columns
including the lower troposphere (TrOC, surface-450hPa, surface-300hPa) derived from the IASI-O3 KOPRA product. This
drift is rather similar for the different columns and about $-0.06 \pm 0.02$ DU/yr, but more pronounced in summer than in winter.
It should be noted that this mean drift is largely dependent on the chosen sample of ozone sonde sites and is heavily dominated
by central Europe. It should then be considered in caution if used to correct IASI timeseries on larger domains.

As in other satellite and ground-based datasets, we found negative tropospheric ozone column anomalies in the 2020-2022
(post-COVID19) time period in our NH mid-latitude domains. These negative ozone anomalies are ascribed to the decreasing
ozone precursor emissions and are strongest in NH midlatitude spring and summer seasons, but are still continuing today (e.g.
Blunden and Boyer, 2024). These may have an impact on the tropospheric ozone trends. The upper tropospheric trends derived
from the IASI-O3 KOPRA product change from positive (moderate uncertainties and small p-values) to more undefined (large
uncertainties and large p-values) trends when the 2020-2022 period is included in the trend calculation period. On the contrary,
the lower tropospheric trends are almost systematically negative in the lower troposphere regardless of whether the 2020-2022
period is included or not. We showed that the trends estimated for the TrOC result from a compensation between the lower
and upper tropospheric behavior. Usually, no specific trend is estimated for TrOC.

In this context of uncertain trends and opposite behavior in the lower and upper troposphere which likely compensate for the
TrOC, the questions about possible drifts, more pronounced in summertime, between our sample of ozone sonde time series
and the IASI retrievals should be investigated in more detail. An extension of the sample with more homogenized NH



midlatitude ozone sonde time series from the HEGIFTOM dataset and other ground-based data sources such as lidar is envisioned.

## Appendices

### Appendix A: Trends and drift calculation

Trend calculations are based on the monthly $O_3$ anomalies. First, we consider the monthly means of $O_3$ partial columns gridded on a 1°x1° resolution grid for each of the three domains. The gridded monthly means may be averaged over subregions corresponding to the IPCC regions (Iturbide et al., 2020). We use the regionmask package for Python to define the regions of interest for this study (https://regionmask.readthedocs.io/en/stable/defined_scientific.html, last access 16 December 2024). The trends are calculated following the recommendations from the TOAR-II activities (Chang et al., 2023) and based on the

quantile regression method. We choose the 50[th] percentile, which represent the median regression. We use the toarstats package for Python (https://gitlab.jsc.fz-juelich.de/esde/toar-public/toarstats, last access 16 December 2024) where the method is implemented. The monthly timeseries are deseasonalized by fitting a sine-cosine combination with periodicities of 12 and 6 months. The quantile regression is then performed, and a moving block bootstrap algorithm is applied to estimate the uncertainties of the derive trends.

The drift is calculated similarly. Instead of 1°x1° gridded monthly timeseries, we averaged the pixels matching the coincidence criteria for the sonde comparison and calculate the monthly means for the sonde and IASI columns. We then calculate the difference between the monthly timeseries of the sonde (raw or smoothed) and IASI and apply the quantile regression (50[th] percentile) similarly to the trend calculation to estimate the drift and its uncertainties.





## Appendix B: Consistency between IASI-A, IASI-B, and IASI-C

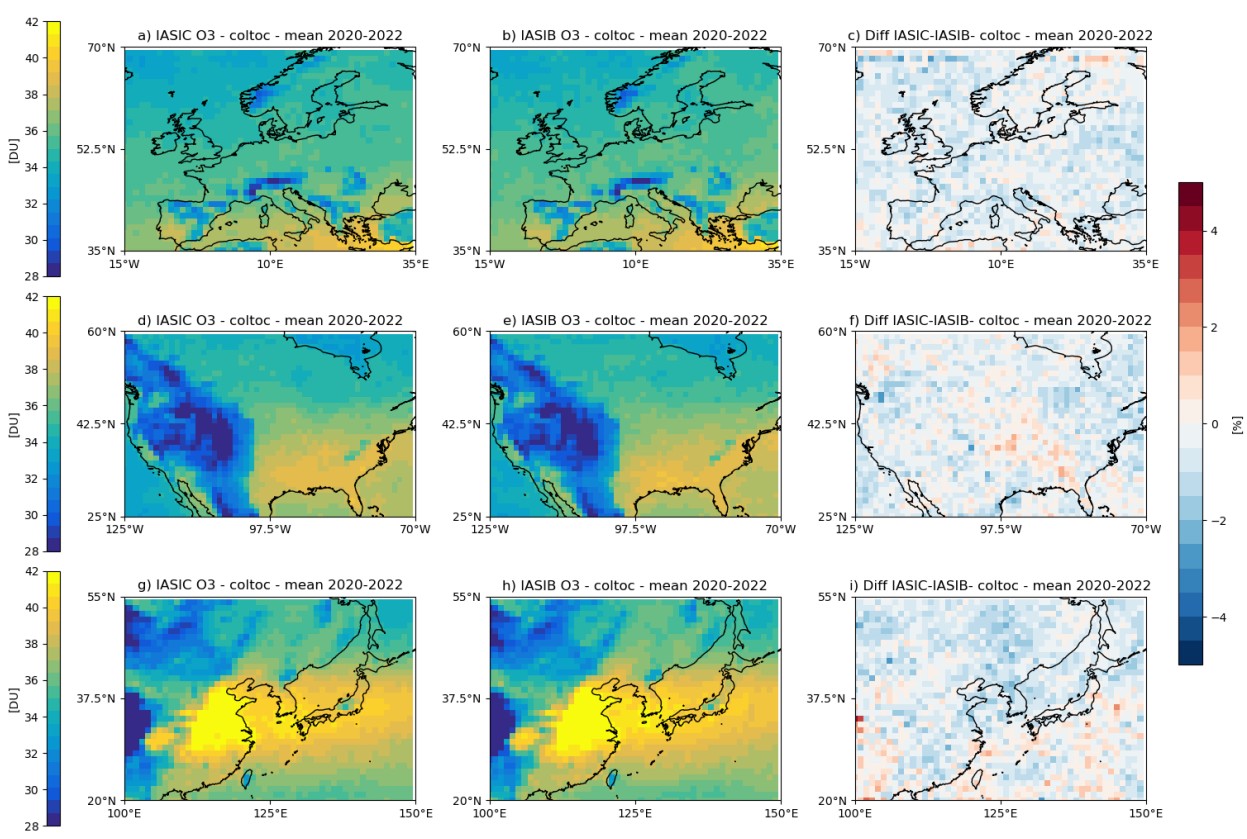


**Figure B1: same as Fig 2 but for IASI-C and IASI-B**

## Appendix C: Global statistics on columns

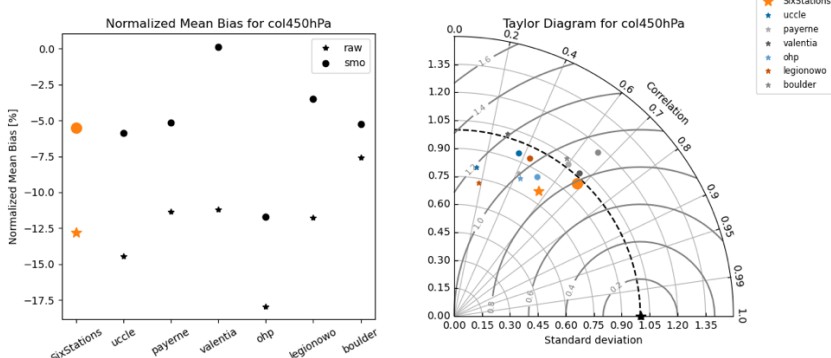

**Figure C1: same as Fig. 6 but for the surface-450hPa partial column**



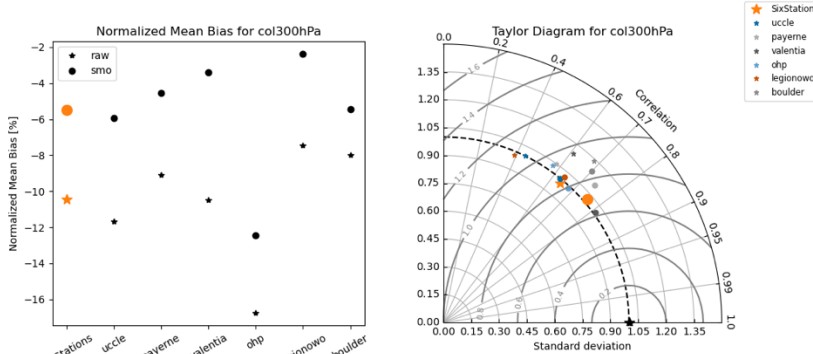

**Figure C2: same as Fig. 6 but for the surface-300hPa partial column**

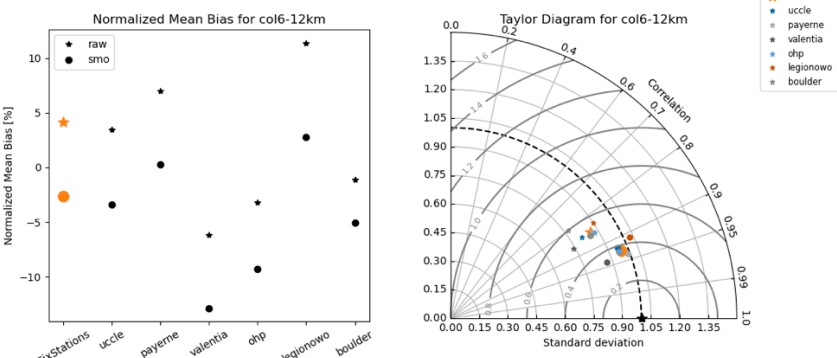

**Figure C3: same as Fig. 6 but for the 6-12km partial column**

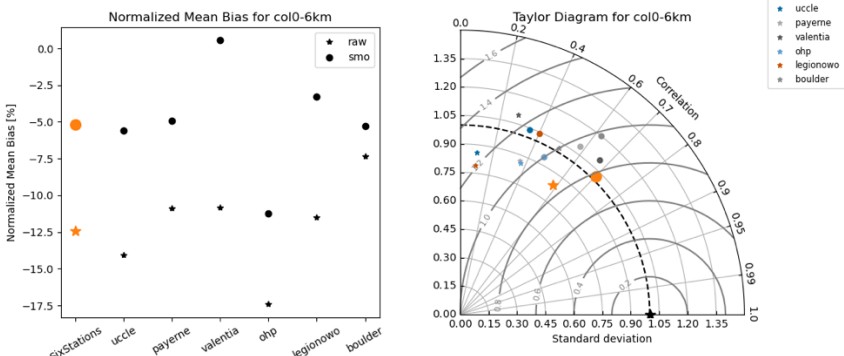

**Figure C4: same as Fig. 6 but for the surface-6km partial column**



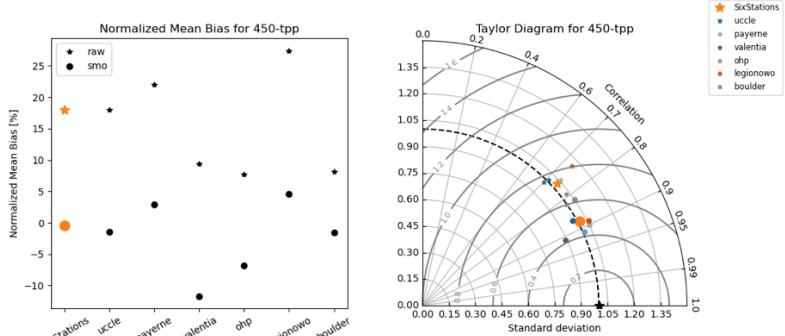

**Figure C5: same as Fig. 6 but for the 450hPa-tropopause partial column**

## Appendix D: Trend estimations

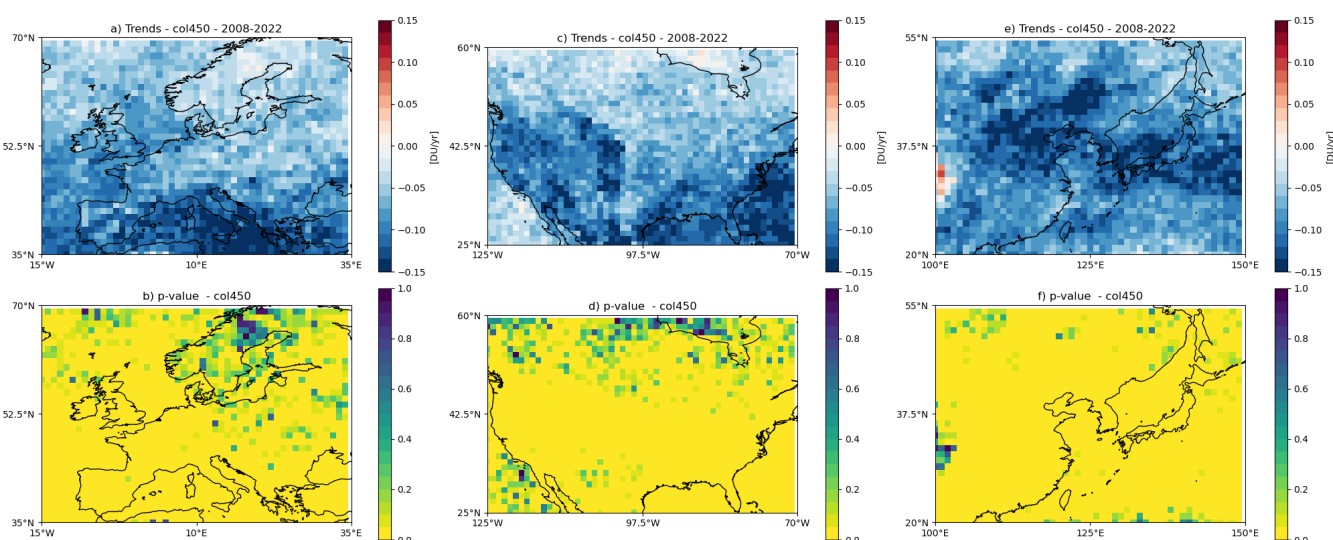

**Figure D1: same as Fig. 9 for the surface-450hPa partial column**





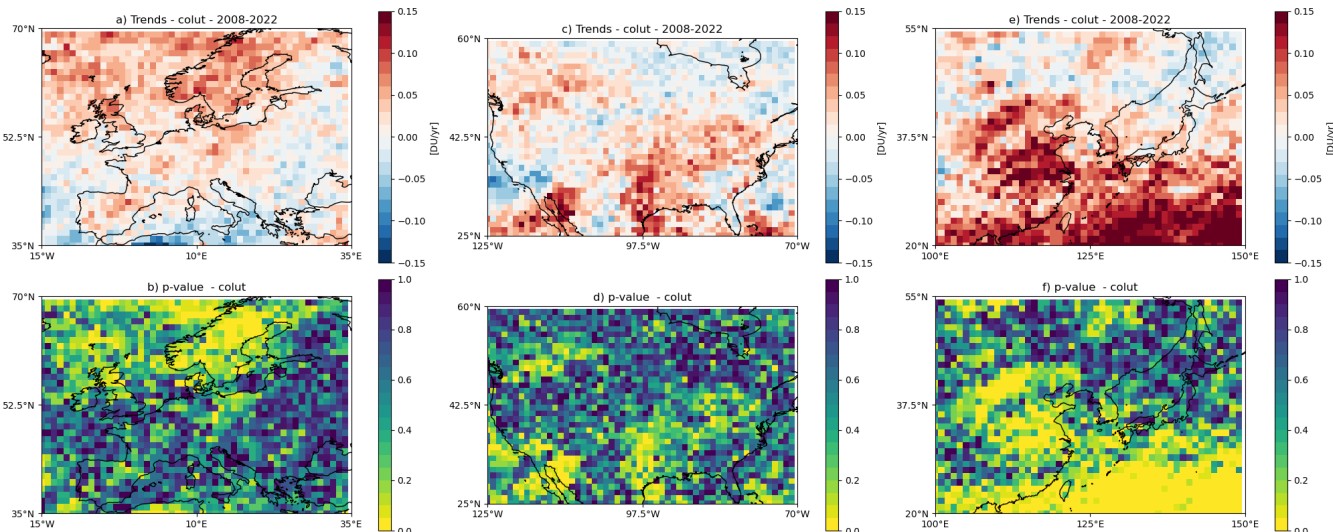

**Figure D2: same as Fig. 9 for the 450hPa-tropopause partial column**

**Table 3: Trends between 2008 and 2019, in DU/yr, calculated for different regions defined for the Sixth IPCC assessment Report and the three main domains. The corresponding p-values are indicated in parenthesis. The trends are provided for three partial columns: TrOC, surface-450hPa, and 450hPa-tropopause. The trend values are bolded when the p-value is smaller or equal to 0.05.**

| Trends | TrOC | 450 hPa | 450hPa-tpp |
|---|---|---|---|
| | DU/yr | DU/yr | |
| **Europe** | +0.02 ± 0.03 (p=0.54) | **−0.06 ± 0.01 (p<0.001)** | **+0.09 ± 0.09 (p=0.003)** |
| WCE | +0.06 ± 0.04 (p=0.13) | **−0.06 ± 0.02 (p=0.02)** | **+0.11 ± 0.05 (p=0.04)** |
| NEU | **+0.09 ± 0.04 (p=0.04)** | −0.02 ± 0.02 (p=0.29) | **+0.10 ± 0.04 (p=0.02)** |
| MED | +0.01 ± 0.05 (p=0.90) | **−0.13 ± 0.02 (p<0.001)** | **+0.10 ± 0.05 (p=0.04)** |
| **North America** | +0.00 ± 0.05 (p=0.99) | **−0.08 ± 0.01 (p<0.001)** | **+0.05 ± 0.03 (p=0.05)** |
| WNA | +0.00 ± 0.05 (p=0.98) | **−0.09 ± 0.02 (p<0.001)** | +0.07 ± 0.04 (p=0.12) |
| CNA | +0.01 ± 0.05 (p=0.84) | **−0.08 ± 0.01 (p<0.001)** | **+0.09 ± 0.04 (p=0.01)** |
| ENA | +0.03 ± 0.05 (p=0.52) | **−0.09 ± 0.02 (p<0.001)** | **+0.11 ± 0.05 (p=0.02)** |
| NCA | −0.02 ± 0.05 (p=0.63) | −0.07 ± 0.05 (p=0.12) | **+0.13 ± 0.04 (p<0.001)** |
| **East Asia*** | | | |
| EAS | **+0.14 ± 0.04 (p=0.001)** | **−0.07 ± 0.02 (p<0.001)** | **+0.14 ± 0.03 (p=0.001)** |


**Code and data availability**

We use the toarstats package for Python (https://gitlab.jsc.fz-juelich.de/esde/toar-public/toarstats, last access: 16 December 2024) to calculate the trends and drifts. We use the regionmask package for Python, which provides the last IPCC region



definition (https://regionmask.readthedocs.io/en/stable/defined_scientific.html, last access: 16 December 2024). The 1°x1°

monthly ozone distributions over Europe, North America and East Asia for TrOC, surface-450hPa, surface-300hPa, surface-6km, and 6-12km partial columns are available on the EaSy Data repository (doi attribution in progress). The homogenized ozone sondes are available from the HEGIFTOM website (https://hegiftom.meteo.be, last access: 16 December 2024). The tropopause information is from the Reanalysis Tropopause Data Repository (Hoffmann and Spang, 2022b).

**Author contribution**

GD managed the study from its conception, the analysis of data, the preparation of the paper, and the funding acquisition. ME provided the IASI-O3 data and contributed to the analyses of the data. RVM, GA, MG, EMB provided the ozone sondes measurements. All authors contributed to the discussion and improvement of the paper.

**Competing interests**

RVM is a member of the editorial board of AMT.

**Acknowledgements**

The IASI mission is a joint mission of EUMETSAT and the Centre National d'Etudes Spatiales (CNES, France). This study has been financially supported by the French Space Agency – CNES (grant no. IASI/TOSCA and TOTICE/TOSCA). The authors acknowledge the AERIS data infrastructure (https://www.aeris-data. fr, last access: 19 October 2021) for providing

access to the IASI level 1C data, distributed in near-real time by EUMETSAT through the EUMETCast system distribution. We acknowledge the Karlsruhe Institute of Technology Institut für Meteorologie und Klimaforschung (KIT-IMK), Karlsruhe, Germany, for a license to use the KOPRA radiative transfer model. This work was granted access to the HPC resources of TGCC under the allocation 2023-A0130107232  made by GENCI.

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
