# Peer review of "Performance assessment of the IASI-O3 KOPRA product for observing midlatitude tropospheric ozone evolution for 15 years: validation with ozone sondes and consistency of the three IASI instruments"

_EGUsphere, 2024_

## Author Response (AR1)

The authors thank the referee for his/her interest in the manuscript. His/her suggestions, recommendations and remarks were very useful for improving the manuscript. In the following referee's comments are indicated in italics and grey color and the reply to each comment is given just below.

*General comments:*
*This manuscript by Dufour et al. provides an insightful performance assessment of the regional KOPRA ozone profile v3.0 product that deserves publication in AMT. The current presentation, however, could be improved regarding scientific clarity and significance. Especially the 'improvements' in this version 3.0 with respect to the previous, the retrieval information as captured by the averaging kernels, and the discrepancies in the drift studies are unclear.*

*Specific comments:*
*Section 2.1 (1): Briefly explain the main differences of this v3.0 of the KOPRA product with respect to previous versions, especially as the end of Section 3.3.2 mentions that its performances are 'similar'.*
The only difference between v3.0 of the KOPRA product and previous versions is that we fit simultaneously water vapor with ozone to account for spectral interferences during the retrieval. It is worth noting that this concerns only residual interferences as a selection of micro windows excluding the main water vapor lines has been done since the first version of the product (Eremenko et al., 2008). We do not expect a large impact on the performances of the product. In Dufour et al. (2018), we reported the main improvement is in the upper troposphere – lower troposphere columns, which do not consider here as we focus only on the troposphere.
We propose to change the text line 91 as follows and not to mention any expected improvements: "**Compared to previous versions of the product**, water vapor is fitted simultaneously with ozone to account for **residual** interferences in the spectral windows used for the retrieval in the current version (3.0) of the product."

*Section 2.1 (2): Why are only morning overpass pixels considered? Please explain.*
As our retrieval algorithm is not built to be an operational product with near-real-time capabilities, we decided to focus only on the morning overpasses for which the sensitivity to the lower troposphere is expected to be larger due to larger thermal contrast than for evening overpasses.
We propose to rephrase the sentence line 98 as follow: "The retrieval is performed for the morning pixels, **when the thermal contrast and then the sensitivity are the largest, and for** three geographical regions…"

*Line 138: The observation that "the differences can vary at the scale of month or year" does not seem to be a sufficient condition for "preventing assessing a systematic bias between the instruments." Please explain and rephrase.*
We rephrased the sentence like this: "The timeseries of the differences between IASI-A, IASI-C, and IASI-B (right panels of Fig. 1) show that the differences have large variabilities: the standard deviations of the mean differences are close to or larger than the mean differences themselves. This prevents assessing a systematic bias between the instruments." and we indicated the mean differences and their standard deviation in Fig 1.

*Lines 180-183: If the regular smoothing equation is used, just cutting the averaging kernel matrix above the sonde vertical range is equivalent to first extending the sonde profile with the a-priori profile and smoothing over the full vertical range afterwards.*
We agree and propose to replace "we need to complete the sonde profile" by "we complete the sonde profile" to avoid any confusion.

*Lines 184-185: "to avoid possible interpolation issues and uncertainties" is quite vague. Is this to conserve the vertically integrated ozone profile or retrieval sensitivity, more specifically?*
Yes, this is to conserve the vertically integrated ozone profile. We then rephrase the sentence as "To conserve the vertically integrated ozone profile, we convert…"

*Line 201: I tend to disagree with the statement that "Outside of this [9-15 km] range, the bias is rather constant in altitude" also given that variations are discussed below. Possibly rephrase?*
Compared to the 9-15 km range, the bias is rather constant between 1 and 7 km. We propose to rephrase as follow: "On the contrary, the bias is rather constant in altitude in the free troposphere between 1 and 7 km where a negative bias …".

*Line 202-203: "When the sonde profile is smoothed by the AKs, the smoothing errors are removed" This is a brief and rather vague statement for an operation with quite some impact. Better elaborate on this in particular, and on the retrieval sensitivity in general.*

Validation of vertical profiles retrieved from satellite observation has a long history now with the Rodgers formalism published in 2000. We are not sure it is necessary to come back in details on this formalism when it is just an application of it without further methodological development. We propose to refer the reader to Rodgers (2000) to help clarify the sentence. In addition, we provide now information on the retrieval sensitivity and errors at the end of section 2.1 as follow, and we included a new figure: "The sensitivity of the retrievals is usually given by the averaging kernels and the degrees of freedom (DOF), which gives an estimate of the number of independent pieces of information in the retrieval (Rodgers, 2000). In the troposphere, the DOF of the TrOC estimated by IASI-O3 KOPRA are about 0.85 on average but they can range from 0.12 to 1.82 depending on the season and the location. Figure 1 shows the distributions of the DOF for the different regions and IASI instruments. The largest DOF values occur in summer in the southern locations of our domains. The DOF distributions of the three IASI instruments are very similar and no significant changes are observed over time. The errors estimated on the TrOC usually range from 15 % to 20 % in summer and from 20 % to 25 % in winter. These degrees of freedom and errors are comparable to other IASI products (Barret et al., 2020, Boynard et al., 2018, 2025)."

*Line 206: An obvious validation research question is whether the observed uncertainties match the (prognostic) errors provided within the KOPRA product files?*

As mentioned previously, additional information has been added on the retrieval sensitivity and errors at the end of section 2.1. At line 206, the following sentence has been added: "The RMSEs are consistent with the retrieval errors, which range from 15% to 30% in the troposphere depending on the altitude and the season."

*Figures 4 and 5 should have axis legends and units. Moreover, red and blue colors are mostly saturated in Figure 5, so an extension of the color scale range might be appropriate.*

Figures 4 and 5 have been revised according to referee's suggestions.

*Lines 267-268: "One can also notice that the variability of the TrOC is slightly underestimated by IASI (by about 10%)" Explain how this can be seen from Figure 6. Also explain all aspects of the Taylor diagram in the latter. It is not clear which "curved lines" are referred to in the caption (light or dark grey).*

The standard deviation gives an estimate of the variability of the TrOC. If the variability of the sondes and the retrieval are similar, the symbol on the Taylor diagram should be on the dashed curve corresponding to a ratio of these variabilities equal to one. In our case, the symbols are rather around the 0.9 curve (light grey) meaning that the variability is underestimated by 10%. We propose to rephrase the sentence as follow: "One can also notice that **all the points range around the 0.9 variability curve of the Taylor diagram (Fig. 6)** meaning the variability of the TrOC is slightly underestimated by IASI (by about 10%)."

The Taylor diagram is now explained in the caption of Fig. 6 as follow: "The light grey curved lines denote the variability given by the standard deviation, the radius the correlation, and the dark grey curved lines the RMSE. Note that the standard deviation and the RMSE are normalized against the standard deviation of the sondes. The black dashed line corresponds to a normalized standard deviation of one, meaning the standard deviation of the retrieval and the sonde are equal. The black star represents the ideal case where retrievals and sondes are in perfect agreement."

*Line 364: "The negative trends reported in the lower troposphere are then not affected by the a priori." This only holds if the retrieval sensitivities do not significantly change either. Can this be ruled out?*

The mean degrees of freedom of the retrieval in 2008 and 2022 are 0.74 and 0.77 respectively for Europe for instance. The small changes between the beginning and the end of the period remain very small (<4%) and likely not significant enough to drive the trends in the troposphere. We rephrased the sentence like this: "As the retrieval sensitivity does not significantly change between the beginning and the end of the period, the negative trends reported in the lower troposphere are then not affected by the a priori".

*Lines 365-369: Possibly quantify the contribution of drift to the trends?*

A table (Table D1) similar to Table 3 but limited to TrOC and surface-450 hPa has been added in Appendix D.

*Line 390: The aim to "provide recommendations for its use in trend analysis" does not appear to me in the text. Does this refer to looking at several tropospheric sub-columns?*

The aim was to provide some tips and warnings for its use in trend analysis, such as (i) the consistency between the three instruments which allows the combination of IASI-A and IASI-B to study longer timeseries, (ii) the

warning concerning a possible drift but difficult to correct. We propose to remove this part of the sentence to avoid confusion to the reader.

*The end of Section 3.3.3 rightfully questions the representativeness of the six ozone sonde station data for the three regional studies in this work, especially regarding long-term drifts. The conclusion in line 407, referring to the fact that the mean drift (properly assessed) should indeed not be "largely dependent on the sample of ozone sonde sites", should be taken at face value and result in a (future) reconsideration of the drift study?*

We restricted the comparison to the homogenized sondes data within our three domains and which covered the entire time period (2008-2022) when we collected the data at the beginning of our study. Unfortunately, at that time, several sonde sites were not available for the recent years, and we excluded them. In the future, we may reconsider the drift study including more homogenized sondes when they will be all available to get a more robust drift assessment as stated at the end of the conclusion. We may also consider performing the KOPRA retrieval outside the three regions around the available sites to have a larger picture of the drift. However, we will always face the question of the representativeness of the sonde sites in regard of the satellite and vice versa as it is shown by the variability of the drifts derived from one site to another.

*It seems more appropriate to keep the brief yet important Appendices A and B with the main text?*

We prefer to keep Appendix A so as not to disrupt the flow of the discussion in Section 2.2 when it is first referred to. We also prefer to keep Appendix B to limit the number of large figures in the main text. The figures are provided as appendices, so easily accessible for the reader.

*Technical corrections:*
*Lines 18-19: repetition of IASI-B*
Corrected
*Line 51: "remain"*
Done
*Line 59: "used to study trends"*
Done
*Line 79: "discusses"*
Done
*Line 85: Explain "L1C" or leave out.*
We leave it out.
*Lines 89 and 92: Specify which a priori information is discussed, differentiating between temperature and ozone profiles.*
Done
*Line 116: Provide latitude-longitude definitions for the regions under study.*
It was already done line 99.
*Line 219: "to calculate"*
Done
*Line 244: Either "using" or "over" but not both.*
Corrected
*Line 255: "errors are removed"*
Done
*Line 304: "spectral fit"*
Done
*Table 2: The last row of data (on 300hPa-tropopause columns) seems to be missing.*
The missing row has been added
*Figure 8: Match the vertical scales of both plots for better comparability?*
Done
*Line 434: "derived trends"*
Done
*Table 3 should be renamed D1 (or similar according to which appendices are kept).*
Done

**RC2**: 'Comment on egusphere-2024-4096', Alistair Bell, 06 Feb 2025  reply
**General Comments**

The authors thank the referee for his interest in the manuscript. His suggestions, recommendations and remarks were very useful for improving the manuscript. In the following referee's comments are indicated in italics and grey color and the reply to each comment is given just below.

*The paper shows a validation of retrievals using the KOPRA algorithm from the three iterations of the IASI instrument, which show a low bias between the partial ozone columns for different altitude layers of the troposphere.*

*The paper also provides analysis of the IASI KOPRA product compared to in-situ sonde measurements. The results show that when sonde measurements are convolved with the KOPRA averaging kernel, the bias is close to zero, and there is an RMSE of around 20% when compared to the KOPRA retrievals. Drifts in retrieved measurements are also compared to ozonesonde measurments, showing a small negative drift, with a low p-value for the tropospheric ozone column.*

*Ozone trends are then analysed for three regions in the northern hemisphere: North America, Europe and East Asia. It is interesting to see that for the lower troposphere, trends are mainly negative for all regions, with mainly low p-values, whereas for the upper troposphere there is little clear signal except in the East China Sea/Pacific region, where trends are positive. It is also interesting to note that the negative trend in the lower tropsophere remains even when the covid period is excluded from the dataset.*

*The article presents important results relevant for tropospheric ozone analysis, and should be published in AMT with corrections and clarifications noted below.*

**Specific Comments**
*It would be useful to specify more precisely any instrumental changes between IASI A, B and C. If there are no instrumental changes at all, this would also be worth highlighting. In case the performance of the instruments is slightly different, it may also be worth plotting the differences in measuremnent response of retrievals from the three instruments.*

The three IASI instruments are completely identical. We added the information line 62: "Three **identical** versions of the instrument". For the periods used for this study, the three instruments were functioning nominally and within their specifications. The performances of IASI instruments are remarkably stable and make IASI a reference instrument, in particular in radiometric terms. We refer the referee to the presentations given during the last IASI conference in 2024. To our knowledge, no publication which addresses this point for the three instruments exists.

*It is intriguing that both IASI-A and -C have roughly the same bias compared to IASI-B for all partial columns. Could you specify the overpass time separation of the satellites? As the rate of ozone formation and depletion could be quite high at the local time for which observations are made, I wonder if the time separation of the satellite could explain this bias?*

The overpass time difference between the satellites is about 45 min. Within our domains, for two different locations, the order in which the locations are sampled by the satellites can be different. If one takes the example of July 1$^{st}$, 2022, Greece is sampled first by IASI-C at about 8:00 UTC (9:30 LT) and then 45 min later by IASI-B (8:45 UTC, 10:15 LT) but Corsica is sampled first by IASI-B at about 8:45 UTC (9:30 LT) and then 45 min later by IASI-C (9:30 UTC, 10:15 LT). It can be different another day depending on the relative configuration of the swaths. It is then not straightforward to link the bias with the time separation of the satellite. The figure below shows the overpass time of IASI-B (left) and IASI-C (right) for July 1$^{st}$, 2022.

[Figure]

*line 175: "The coincidence criteria used for the validation are ±1° in latitude and ±1° in longitude around the sonde station, a time175 difference shorter than ± 6 hours" - can you confirm that there was no systematic change in the time difference between iasi/sonde observations?*

There is no systematic change with time in the time difference for each individual site. It is worth noting that from one site to another the time difference can be different but remain similar over the period.

*Section 3.3.1 It would be useful to compare these results with other IASI ozone retrievals, such as those from Boynard et al. (2016, 2018), to note whether the biases and RMSE values observed in here are consistent with other retrieval algorithms*

We provide, now, some comparisons with the SOFRID-O3 v3.5 (Barret et al., 2020) and the IASI-CDR O$_3$ (Boynard et al., 2025) products. We added this paragraph after line 210: "The IASI-O3 KOPRA product has very similar negative bias in the free troposphere compared to the SOFRID-O3 v3.5 product bias, which ranges between -10 % and -5 % (Barret et al., 2020). In the upper troposphere lower stratosphere (UTLS), the biases around 15 % for smoothed profiles are also in good agreement. Compared to the IASI-CDR O$_3$ product (Boynard et al., 2025), a similar agreement is observed in the UTLS. In the free troposphere, the IASI-CDR O3 product seems to show a larger negative bias (between 15-20%) compared to IASI-O3 KOPRA and SOFRID-O3 v3.5." We added also this statement after line 249: "As for profiles, the mean normalized bias for IASI-O3 KOPRA TrOC is similar to the ones reported for SOFRID-O3 v3.5 (Barret et al., 2020) for smoothed sondes in the northern midlatitudes (about - 3 %) and smaller than the one reported for the IASI-CDR O3 product (Boynard et al., 2025) in the midlatitudes (about -10 %). It is worth noting that the set of sonde sites considered for the validation of these different products is different and may explain some of the differences."

*line 208: "The largest differences and RMSE in the first two kilometers when comparing to raw sondes are likely due to an issue because high altitude stations (Payerne, OHP, and Boulder) are mixed with low altitude stations."*
*- It is not clear to me why this would create a large bias and rmse, more likely your next point about the lack of sensitivity of satellite observations near the surface.*

We finally find a small issue on how the first altitudes were managed for altitude sites. It has been fixed, and the corrected figure is now provided in the revised manuscript. We then remove lines 208-211 in the revised manuscript.

*figure 5: This is a nice plot, but seems to be aiming to show different things that could perhaps be better expressed with other plots. For example, to show the drift of IASI retrievals, the raw/smoothed sonde retrievals minus IASI retrievals could perhaps be plotted over time. This might more clearly show a drift. Where the negative anomolies are highlighted in 2020 for example, the problem is that from the combination of sonde datasets, it is not clear how many profiles from each location are used at which time. It may change conclusions if anomolies are not uniformly distributed across all locations.*

Figure 5 has been fully revised to improve the contrast in the plot and to add panels with IASI minus raw/sonde to highlight the possible drift. We added also a panel indicated the evolution of the number of profiles for each station. It shows that the contribution of each site is rather stable over time. Lines 235-237 are replaced with: "This effect seems to be more pronounced for IASI till the end of 2022. The contribution from a possible drift in the IASI data cannot be ruled out. Indeed, Fig. 5 (first two rows of the second column) shows the difference between IASI and sondes anomalies. In the free troposphere, this difference is negative and slightly increases with time. This will be analyzed in more detail with the partial columns in the next sections. It is worth noting that mean sonde profiles are mainly driven by Payerne and Uccle sites, and that the relative contribution of each site is rather constant over time (Fig. 5, bottom right panel)." The new Figure 5 looks as follows:

[Figure]

[Figure]

*Figure 9: Interesting to see that the trends in the Mediteranean sea, Bay of Biscay and more Southernly part of the North Atlantic Ocean all have negative trends with low P-values, whilst the East China Sea/Pacific have postive trends with low P-values. Are there changes in ozone precursors that could have caused this?*

It is out of the scope this validation paper to assess the causes of the trends. We can mention here that the emissions of $O_3$ precursors, especially NOx, have been slowly decreasing since more than 20 years in Europe. It may influence the negative trend observed in southern Europe depending on the regional chemical regimes. In China, the NOx emissions have started to decrease in the mid-2010s leading to an increase of surface ozone since 2015. However, the positive trend in East Asia is mainly visible in the upper troposphere where the influence of surface emissions is likely small. This would deserve a dedicated study.

**Technical Corrections**

*line 17: Please define KOPRA before first usage*
Done
*line 20: please correct to make clear: "across the three study domains: Europe, North America, and East Asia." (or similar)*
Done
*line 38: ...in addition to **being**...*
Corrected
*line 55: **were** more likely negative*
Corrected
*line 59: ...for trend studies...*
Corrected
*line 95: lower than?*
Corrected
*Figure 3: I found it quite hard to find and guage the size of some circles on this plot - I wonder if it meets requirements for visual impairments. Perhaps it would be better to remove the topographic shading?*
The figure has been revised. We added red edge color circle and provided the number of days with observations for each site in the caption.
*Figure 4: Plots require axis labels and units*
Done
*Figure 5: again no y axis label*
Done
*Figure 6 (right): I'm not sure if I see the Boulder star. Maybe you can add more colours to the plot to distinguish stations?*
We changed the colors. Concerning Boulder, the raw and smoothed statistics for the Taylor diagram are the same. The star is under the circle and then not visible. We added a note in the caption.
*Figure 8: is TOC the same as TrOC?*
Yes, it is. We harmonized the figure with text and change TOC to TrOC.
*line 419: "In this context of uncertain trends and opposite behavior in the lower and upper troposphere which likely compensate for the TrOC, the questions about possible drifts, more pronounced in summertime, between our sample of ozone sonde time series and the IASI retrievals should be investigated in more detail." - this sentence should be rephrased or broken down into two sentences.*
The sentence has been rephrased as follow: "The questions raised by our study regarding possible drifts, mainly in summer, between ozone sondes and IASI retrievals, as well as the representativity of the sondes for their correction, should be investigated in more detail in the future."

**RC3**: 'Comment on egusphere-2024-4096', Anonymous Referee #3, 14 Feb 2025  reply

The authors thank the referee for his/her interest in the manuscript. His/her suggestions, recommendations and remarks were very useful for improving the manuscript. In the following referee's comments are indicated in italics and grey color and the reply to each comment is given just below.

*The study by Dufour et al. provides a useful assessment of long-trends from the IASI-O3 KORPA product. The general outline of the paper is straight forward and provides some useful additions to the literature. For instance,*
*- a useful assessment of the consistency of the IASI-A, IASI-B and IASI-C products (i.e. IASI-B as reference and biases within 1%).*
*- a useful, though limited frequency, comparison of IASI-O3 with ozonesondes in Europe and the US.*
*- derivation of a useful drift trends in multiple sub-columns of the data.*
*- analysis of the IASI-O3 trends for multiple sub-columns (and testing the significance of the COVID-19 years). Therefore, this manuscript is suitable for publication in AMT subject to some minor comments.*

*Minor Comments:*
*1) The last sentence of the abstract is unclear. Please reword.*
We reword the sentence as follow: "The negative tropospheric ozone column anomalies observed in 2020-2022 (post-COVID19 period) only slightly impact the trends already on-going for 2008-2019."
*2) Line 42 - Where you say "relatively short lifetime", please provide and example. For reactive trace gases, one could say O3 has a long lifetime given the lifetime of e.g. OH and NO2. I think this can be put more into context (i.e. the O3 lifetime).*
We propose to rephrase as follow: "With a relatively short lifetime (days to weeks) compared to other greenhouse gases"
*3) Line 87 - Are there issues with how the Dufour 2012 and 2015 references have been written?*
It has been corrected.
*4) Line 106 - Why not include IASI-C?*
We wanted to provide trends on a continuous period from the beginning of the IASI period (2008) and 2022. We then choose to combine IASI-A and IASI-B. We do not consider IASI-C as the 2019 is not entirely sampled. We show in the paper a good consistency between the three instruments, so this choice should not impact the estimated trends outside their uncertainties.
*5) Line 111: Could the authors add some information on the DOFS of the different sub-columns? This might help put the ability of IASI to retrieve O3 at different levels in perspective. An additional table with the information in for the 3 regions of interest could be useful.*
We provide now information on the retrieval sensitivity and errors at the end of section 2.1, as follows and we included a new figure: "The sensitivity of the retrievals is usually given by the averaging kernels and the degrees of freedom (DOF), which gives an estimate of the number of independent pieces of information in the retrieval (Rodgers, 2000). In the troposphere, the DOF of the TrOC estimated by IASI-O3 KOPRA are about 0.85 on average but they can range from 0.12 to 1.82 depending on the season and the location. Figure 1 shows the distributions of the DOF for the different regions and IASI instruments. The largest DOF values occur in summer in the southern locations of our domains. The DOF distributions of the three IASI instruments are very similar and no significant changes are observed over time. The errors estimated on the TrOC usually range from 15 % to 20 % in summer and from 20 % to 25 % in winter. These degrees of freedom and errors are comparable to other IASI products (Barret et al., 2020, Boynard et al., 2018, 2025)."
*6) Line 118: Boynard et al reference format needs correcting.*
It has been corrected.
*7) Line 116 - Should be "IASI-C" and should be an "a" between "have" and "common".*
It has been corrected.
*8) Line 123: Need "is" between "deviation" and "smaller".*
It has been corrected.
*9) Figure 1, what does "coltoc" mean? Table 1, need a full stop at end of caption sentence.*
We changed coltoc to TrOC for clarity.
*10) Figure 2 - Please write IASIA as IASI-A in titles of figure panels etc., to make it clearer.*
Done
*11) Line 167 - You use the Boulder ozone site, but due to the orography around the site, could that adversely impact on the ability of the satellite to make sensible retrievals to be compared with the sonde?*
The sonde profiles are re-grided (in the partial columns space) on the IASI vertical grid. Sondes and retrievals then share the same surface altitude for column calculation, this should limit the orography effect in the comparison of the column. For the profiles given in volume mixing ratio in Fig. 4, we corrected a small issue on how the first altitudes were managed for altitude sites and the calculation of averaged profile.
*12) Figure 4: Colours chosen are not overly clear for the vertical profiles and the legends are not overly clear either in telling the reader what they are representing.*
The figure has been improved.
*13) Figure 5: Might have missed this, but for which spatial region is used for this figure time-series?*

The monthly anomalies are computed from the coincident profiles of each sonde site. It does not correspond to one of our three regions. Figure 5 has been revised according to other referees' comments and the fact the averaged is computed at the sonde sites is clarified in the caption.

*14) Line 225: Should this be "RMSE", not "RSME"?*

It has been corrected.

*15) Figure 7: Can the labels/legends on the RHS of the panels be made bigger and clear?*

The figure has been improved.

*16) Lines 310-312: This sentence is not overly clear, especially the term "the AKs should integrate this loss of sensitivity". Could this be reworded slightly to make it clearly what you are doing?*

The sentence is rephrased as follow, hoping it is clearer: "However, if the change in thermal contrast leads to a loss of sensitivity in the lower troposphere, the same impact should be visible in the smoothed sondes too. Indeed, the smoothing by the AKs transforms the sonde profiles into the same space than the retrieval with similar vertical resolution and sensitivity. As the discrepancy between IASI and the smoothed sonde persists, the change in thermal contrast cannot explain the differences."

*17) In Appendix C, I can see sub-columns for surface to 6 km and surface to 450 hPa. These two regions are very similar, so what was the motivation for using both sub-column ranges in the analysis?*

We agree that the two sub-columns are very similar. We chose to present the surface to 6 km to remain comparable with the sub-columns in previous study (e.g. Dufour et al, 2012). We present surface to 450 hPa to be comparable to other studies of the TOAR-II (e.g. Pope et al., 2024).

**RC4**: 'Comment on egusphere-2024-4096', Anonymous Referee #4, 23 Feb 2025 reply

The authors thank the referee for his/her interest in the manuscript. His/her suggestions, recommendations and remarks were very useful for improving the manuscript. In the following referee's comments are indicated in italics and grey color and the reply to each comment is given just below.

*General Comments*

*This manuscript, titled "Performance assessment of the IASI-O3 KOPRA product for observing midlatitude tropospheric ozone evolution for 15 years: validation with ozone sondes and consistency of the three IASI instruments", by Dufour et al. assesses the v3.0 Ozone product from KOPRA. This manuscript can be accepted after major revisions, including some discussion on the improvements made in v3.0 compared to earlier versions and technical corrections.*

*Specific Comments*

- *All figures use small fonts. Please consider increasing the font size.*

All the figures have been revised.

- *Please expand "coltoc" in the plot or define it in the figure caption in Figures 1 and 2.*

"coltoc" has been replaced by "TrOC" to be consistent with the main text.

- *Please use a period at the end of the caption in Table 1.*

Done

- *Please consider outlining the circles in Figure 3 to make them stand out from the background colors.*

The circles have been outlined in red.

- *The font size used in Figure 4 is too small, and the x and y axes don't have labels. The colors chosen are not clear.*

Figure 4 has been revised.

- *Figure 5 is missing the y-axis label, and the colors appear to be saturated in the plot. Please consider extending the colorbar or gridding it into discrete bins.*

Figure 5 has been fully revised to improve the contrast in the plot and to add panels with IASI minus raw/sonde to highlight the possible drift. We also added a panel indicated the evolution of the number of profiles for each station. It shows that the contribution of each site is rather stable over time. Lines 235-237 are replaced with: "This effect seems to be more pronounced for IASI till the end of 2022. The contribution from a possible drift in the IASI data cannot be ruled out. Indeed, Fig. 5 (first two rows of the second column) shows the difference between IASI and sondes anomalies. In the free troposphere, this difference is negative and slightly increases with time. This will be analyzed in more detail with the partial columns in the next sections. It is worth noting that mean sonde profiles are mainly driven by Payerne and Uccle sites and that the relative contribution of each site is rather constant over time (Fig. 5, bottom right panel)." The figure can be found in the reply to referee #2.

- *L98 – Please explain why the morning pixels alone are used.*

As our retrieval algorithm is not built to be an operational product with near-real-time capabilities, we decided to focus only on the morning overpasses for which the sensitivity to the lower troposphere is expected to be larger due to larger thermal contrast than for evening overpasses.

We propose to rephrase the sentence line 98 as follow: "The retrieval is performed for the morning pixels, **when the thermal contrast and then the sensitivity are the largest, and for** three geographical regions…"

- *L206 – Please explain if the uncertainties mentioned agree with those listed in the KOPRA data files.*

Additional information has been added on the retrieval sensitivity and errors at the end of section 2.1. At line 206, the following sentence has been added: "The RMSEs are consistent with the retrieval errors, which range from 15% to 30% in the troposphere depending on the altitude and the season."

- *L123 - What are the other reasons leading to this outcome?*

The biases are rather insignificant compared to their uncertainties. The standard deviation is smaller when IASI-B and IASI-C are compared. We think it could be due to the shorter period of comparison leading to a lower diversity of atmospheric situations encountered compared to the longer period of comparison for IASI-A and IASI-B. We do not see other outcomes to this. We rephrased the lines 123-124 as follows to be clearer: "The bias between IASI-C and IASI-B tends to be larger (but still insignificant) than the one between IASI-A and IASI-B and the standard deviation is smaller. This might be explained by the fact the period of comparison between IASI-B and IASI-C is shorter with probably a lower diversity of atmospheric situations encountered and then smaller standard deviations."

- *Please explain in Section 2.1 how the current v3.0 of KOPRA is better/different compared to older versions.*

The only difference between v3.0 of the KOPRA product is that we fit simultaneously water vapor with ozone to account for spectral interferences during the retrieval. It is worth noting that this concerns only residual interferences as a selection of micro windows excluding the main water vapor lines has been done since the first version of the product (Eremenko et al., 2008). We do not expect a large impact on the performances of the product. In Dufour et al. (2018), we reported the main improvement is in the upper troposphere – lower troposphere columns, which do not consider here as we focus only on the troposphere.

We propose to change the text line 91 as follows and not to mention any expected improvements: "**Compared to previous versions of the product**, water vapor is fitted simultaneously with ozone to account for **residual** interferences in the spectral windows used for the retrieval in the current version (3.0) of the product."

- *L175 – "The coincidence criteria used for the validation are ±1° in latitude and ±1° in longitude around the sonde station, a time difference shorter than ± 6 h, and a minimum of 10 clear-sky IASI pixels matching these criteria." – Please explain if there are any systematic differences observed between the IASI and the sonde data.*

There is no systematic change with time in the time difference for each individual site. It is worth noting that from one site to another the time difference can be different but remain similar over the period.

**Technical Corrections**

- *Abstract – L16 – Metop-B (2012 - Present) and Metop-C (2018-Present) instead Metop-B (2012 - ) and Metop-C (2018-).*

Done

- *Abstract – L17- Please define KOPRA before initial use.*

Done

- *Abstract – L18 and L19 – "The IASI-O₃ KOPRA …" Please check the sentence structure. The sentence is incomplete/has been repeated.*

We replaced "for IASI-A, IASI-B , and IASI-C" by "of the three instruments".

- *Abstract - L19- "IASI-B shows" instead of "IASI-B show"*

Corrected

- *Abstract – L22 through L25 – Please consider splitting the sentence for clarity.*

The sentence has been rephrased as follow: "The comparison with homogenized ozone sondes for six northern midlatitude stations reveals a small negative bias of about 3-6% of the IASI-O3 KOPRA products in the troposphere for both profiles and columns. A rather good correlation between 0.7 and 0.9 is observed and an error of about 15-17% (compared to sondes smoothed with averaging kernels (AKs)) is estimated."

- *Abstract – L22 – Please explain why the O₃ products derived from IASI-A and IASI-B have been used without any bias correction.*

We propose removing this point from the abstract as it is not possible to explain in detail the reason in the abstract. It is clear later in the main text that due to the good consistency between the instruments, we don't need to apply a bias correction.

- *Abstract – L23 – "about 3-6% in" instead of "about 3-6% of"*

Corrected

- *Abstract -L24- "Based on the comparison with the ozone sondes, we identified a temporal drift (of about -0.06 ± 0.02 DU/yr in average), while more pronounced in summer, for three different ozone columns (TrOC, surface-450hPa, surface-300hPa)." Please check the sentence structure, and the grammatical errors.*

The sentence has been rephrased like this: "Based on the comparison with the ozone sondes, we identified a temporal drift (of about -0.06 ± 0.02 DU/yr on average) for three different ozone columns (TrOC, surface-450hPa, surface-300hPa). This drift can be more pronounced in summer."

- *Abstract -L29- Please consider replacing "Whereas" with "While" at the start of the sentence.*

Done

- *Abstract -L31- Remove "the" in "regions the most affected…"*

Done

- *Abstract – In the abstract and throughout the paper, the word "Ozone" is spelled out while it has already been defined as "$O_3$". Please consider using $O_3$ instead of "Ozone" if it has been defined already.*

Ozone is a common noun and not an acronym like KOPRA for example, so we think we can use it throughout the text without disturbing the understanding of the reader.

- *L37 – "due to its decisive.." instead of "chemistry for its decisive role…"*

Corrected

- *L38 – "In addition to be a…" "being" instead of "be"*

Corrected

- *L45 – "evolution is thus crucial" instead of "evolution is then crucial"*

Corrected

- *L47 through L50 – Please consider splitting the sentence for clarity. L50 – "even towards the lowermost.." This part doesn't read smoothly.*

The sentence was split as follows: "In complement to in situ, ground-based, or aircraft measurements, satellite instruments provide a daily global monitoring capability at high resolution with a good sensitivity to probe the troposphere (e.g. Barret et al., 2020; Boynard et al., 2018; Eremenko et al., 2008; Hayashida et al., 2018; Hubert et al., 2020; Liu et al., 2010; Maratt Satheesan et al., 2024; Miles et al., 2015; Pope et al., 2023; Ziemke et al., 2019). Combining ultraviolet and infrared sounders leads to an improved sensitivity towards the lowermost layers (e.g. Cuesta et al., 2013; Fu et al., 2013)."

- *L53 through L57 – Please consider splitting the sentence for clarity.*

Done like this: "Indeed, Gaudel et al. (2018) showed discrepancies especially for the trends derived from these observations. The ones issued from ultraviolet (UV) sounders suggest positive recent trends in tropospheric ozone whereas the ones issued from infrared (IR) sounders were more likely negative."

- *L55 – "Ozone Monitoring Instrument" instead of "OMI Monitoring Instrument"*

Corrected

- *L57 – "such as Ziemke et al. (2019)" instead of "for example (Ziemke et al., 2019)"*

Corrected

- *L62 – "They fly" instead of "They are flying".*

Corrected

- *L70 through L75 – Please check for grammar and sentence structure.*

The sentence was split and rephrased like this: "Trends estimated from satellite observations and reported in literature (e.g. Gaudel et al., 2018, Pope et al., 2024) show some inconsistencies and large uncertainties. This stresses the need for detailed validation of the satellite observations, including the analyses of possible drifts in the timeseries."

- *L85 – Please define L1C.*

We removed it for simplicity.

- *L88 – Please check the Dufour et al. citation format.*

Done

- *L90 – "975-1100 $cm^{-1}$ range" instead of "range 975-1100 $cm^{-1}$ "*

Corrected

- *L92 – Please consider italicizing the words "a priori" here and elsewhere in the paper.*

As far as we know, the term "a priori" is not italicizing in validation papers published in Copernicus journals.

- *L95 – "lower than" instead of "lower to".*

Corrected

- *L96 – "greater than" instead of "larger than".*

Corrected

- *L106 through 107 – Please check for grammatical errors.*

"but" has been replaced by "and".

- *L113 – "long period" instead of "large period", "similar to" instead of "in common with"*

Corrected

- *L113 through 114 – Please consider splitting the sentence for clarity.*

Done like this: "The IASI-B instrument has a long period of operation similar to IASI-A (2013-2018) and with IASI-C (2020-2022). On the contrary, IASI-A and IASI-C do not have a common period of operation with high quality data."

- *L118 – Please check the formatting of the paper cited.*

Done

- *L134 – "a similar drop" instead of "the same drop" ?*

Corrected

- *L148 – Please consider removing "as".*

Done

- *L159 – "HEGIFTOM" – Please abbreviate after expanding.*

Done

- *L162 – "corrected for" instead of "corrected from"*

Corrected

- *L164 – Please consider removing the word "including".*

Done

- *L180 through L185 – "To smooth the sonde profiles by the IASI AKs, we need to complete the sonde profile up to 60 km, the altitude range of the IASI product. As the top altitude of the sonde profiles is variable and as we are mainly interested by the troposphere, we decided to complete the sonde profiles with the a priori profiles from 20 km to 60 km altitude to maintain a certain homogeneity in the sonde treatment. For comparison and AKs smoothing, we need to use the IASI vertical grid (typically 1 km resolution in the troposphere) and regrid the sonde profile to this grid." Please consider rephrasing it to something like, "To smooth the sonde profiles by the IASI AKs, the sonde profile needs to be extended up to 60 km, the altitude range of the IASI product. As the top altitude of the sonde profiles is variable and as we are mainly interested in the troposphere, the sonde profiles were completed/extended with the a priori profiles from 20 km to 60 km altitude to maintain a certain homogeneity in the sonde treatment. For comparison and AKs smoothing, we used the IASI vertical grid (typically 1 km resolution in the troposphere) and re-gridded the sonde profile to this grid."*

We rephrased the sentences as follows: "To smooth the sonde profiles by the IASI AKs, the sonde profile is extended up to 60 km, the altitude range of the IASI product. As the top altitude of the sonde profiles is variable and as we are mainly interested in the troposphere, the sonde profiles were completed with the a priori profiles from 20 km to 60 km altitude to maintain a certain homogeneity in the sonde treatment. For comparison and AKs smoothing, we used the IASI vertical grid (typically 1 km resolution in the troposphere) and re-gridded the sonde profile to this grid."

- *L255 – "RMSE" instead of "RSME"*

Corrected